

# Ethene, propene, butene and isoprene emissions from a ponderosa pine forest measured by Relaxed Eddy Accumulation

Robert C. Rhew[1], Malte Julian Deventer[1], Andrew A. Turnipseed[2], Carsten Warneke[3,4], John Ortega[5], Steve Shen[1], Luis Martinez[6], Abigail Koss[3,4], Brian M. Lerner[3,4,a], Jessica B. Gilman[3,4], James N. Smith[5,b], Alex B. Guenther[7], and Joost A. de Gouw[3]

[1] Department of Geography and Berkeley Atmospheric Sciences Center, University of California Berkeley, Berkeley, CA 94720-4740, USA

[2] 2B Technologies, Boulder CO 80301, USA

[3] Cooperative Institute for Research in Environmental Sciences (CIRES), University of Colorado Boulder, Boulder CO 80309, USA

[4] NOAA Earth System Research Laboratory, Boulder, CO 80305, USA

[5] Atmospheric Chemistry Observations and Modeling (ACOM), National Center for Atmospheric Research, Boulder, CO, 80301, USA

[6] Ernest F. Hollings Undergraduate Scholarship program, NOAA, USA

[7] Department of Earth System Science, University of California Irvine, Irvine, CA 92697-3100, USA

[a] *now at*: Aerodyne Research Inc., Billerica, MA 01821-3976, USA

[b] *now at*: Department of Chemistry, University of California Irvine, Irvine, CA 92697-2025, USA

*Correspondence to*: Robert C. Rhew (rrhew@berkeley.edu)

## Abstract

Alkenes are reactive hydrocarbons that influence local and regional atmospheric chemistry, playing important roles in the photochemical production of tropospheric ozone and in the formation of secondary organic aerosols. The simplest alkene, ethene (ethylene), is a major plant hormone and ripening agent for agricultural commodities. The group of light alkenes ($C_2$-$C_4$) originates from both biogenic and anthropogenic sources, but their biogenic sources are poorly characterized, with limited field-based flux observations. Here we report net ecosystem fluxes of light alkenes and isoprene from a semi-arid ponderosa pine forest in the Rocky Mountains of Colorado, USA using the relaxed eddy accumulation (REA) technique during the summer of 2014. Ethene, propene and butene emissions have significant diurnal cycles, with maximum emissions at midday. The fluxes were correlated with each other and followed general ecosystem trends of $CO_2$, water vapor, light and temperature. The light alkenes contribute significantly to the overall biogenic source of reactive





hydrocarbons, roughly 15 % of the dominant biogenic VOC, 2-methyl-3-buten-2-ol. The measured ecosystem scale fluxes are nearly twice as large as estimates used in global emissions models for this type of ecosystem.

## 1. Introduction

In the troposphere, alkenes contribute to the photochemical production of tropospheric ozone. The "light alkenes",
defined here as the $C_2$-$C_4$ alkenes, include $C_2H_4$ (ethene), $C_3H_6$ (propene) and $C_4H_8$ (1-butene, trans-2-butene, cis-2-butene, and 2-methylpropene). Alkenes are especially important contributors to ozone production in the urban environment where they produce the most ozone per C atom oxidized; ethene and propene have the highest ozone production rates per carbon, followed by isoprene (Chameides et al., 1992;Seinfeld and Pandis, 1998). Like other NMHCs, these alkenes are initially oxidized by the hydroxyl radical (•OH), yielding intermediate peroxy radicals, which oxidize NO to $NO_2$. Oxygen atoms
released in the photodissociation of $NO_2$ can react with $O_2$ to form $O_3$. Other reactions can yield organic nitrates that act as temporary reservoirs and transporters of $NO_x$ (Poisson et al., 2000).

Light alkenes in the atmosphere originate from both anthropogenic and biogenic sources. Ethene, propene and butene are produced industrially by cracking petroleum hydrocarbons, and their double bond makes them versatile chemical feedstocks for industrial reactions. Ethene (also called ethylene) is the most abundant industrially produced organic
compound, with global production capacity in 2009-2011 at 120 to 140 Tg yr$^{-1}$ (Tg= $10^{12}$ g = million metric tonnes) and U.S. production at ~23 Tg yr$^{-1}$ (McCoy et al., 2010;UNEP, 2013). Propene (also known as propylene) is the raw material for polypropylene plastics and other products, and it is the second most abundant organic industrially produced compound, with production rates roughly half of ethene. Currently, global production of ethene and propene is estimated to amount to over 200 Tg per year, or about 30 kilograms per person on Earth (Sholl and Lively, 2016). Anthropogenic emissions are only a
fraction of that at 5.5 Tg yr$^{-1}$ and 2.5 Tg yr$^{-1}$ for ethene and propene, respectively, and mostly emanate from fuel combustion (Poisson et al., 2000). However, leakage of these compounds from industrial areas can impact regional atmospheric chemistry. For example, petrochemical ethene and propene were the primary non-methane hydrocarbons (NMHCs) responsible for high ozone ($O_3$) concentrations near Houston during the 2000 TexAQS study (Wert et al., 2003;Ryerson et al., 2003;de Gouw et al., 2009).

Naturally produced alkenes are believed to be a significant portion of overall carbon contribution of biogenic VOCs (BVOCs) to the atmosphere. Light alkene emissions are roughly 10 % of isoprene (2-methyl-1,3-butadiene, $C_5H_8$), which is the dominant BVOC emitted globally (Poisson et al., 2000;Guenther et al., 2006). However, the spatial and temporal distributions of light alkene emissions are mostly unknown. While hundreds of studies have been conducted on isoprene emissions, including thousands of measurements on leaves, branches and whole plants (Guenther et al., 2006), global
estimates of ethene emissions from plants (11.1-11.8 Tg C yr$^{-1}$) (Poisson et al., 2000;Singh and Zimmerman, 1992) are based largely on one laboratory study (Sawada and Totsuka, 1986), which incorporated 30 sets of incubations of plant shoots from five agricultural plants (wheat, cotton, bean, tomato and orange) and mesquite. These values were then extrapolated to all





vegetation globally, scaled to biomass while omitting species effects, plant growth phase, stress, seasonality, or diurnal trends in emissions.

Biogenic light alkene fluxes have been measured in only a few field studies. Large flux variability was observed in the net ecosystem fluxes of light alkenes at a temperate deciduous forest in Massachusetts (Harvard Forest), measured using a tower-based flux gradient method (Goldstein et al., 1996). Average emission rates at Harvard Forest were similar to the laboratory-based measurements reported by Sawada and Totsuka (1986), which is surprising given the very different measurement conditions and methods. Other studies used flux chambers for surface-atmosphere exchange from low-lying vegetation; studies at a boreal wetland and forest floor in southwest Finland (Hellén et al., 2006) and a rice field in Texas (Redeker et al., 2003) showed that those ecosystems are unlikely to be important sources of light alkenes. Elevated concentrations of alkenes were also observed in the ambient air of tropical forests in Brazil (Zimmerman et al., 1988) and in the upslope air flow in Hawaii (Greenberg et al., 1992), suggesting a local natural source for these compounds. The former was suggested to be largely from biomass burning and the latter from marine emissions, but the potential for biogenic terrestrial emissions was also noted.

The natural abiotic production of light alkenes can also occur through the photochemical processing of dissolved organic carbon in seawater (Ratte et al., 1998;Ratte et al., 1993;Wilson et al., 1970). This process is believed to account for the majority of ethene production from rice fields, as evidenced from control experiment fluxes (Redeker et al., 2003). A separate abiotic production mechanism for ethene and propene has recently been reported from dry leaf litter, with emission rates increasing with temperature (Derendorp et al., 2011). However, these abiotic production rates were estimated to be insignificant in their global budgets.

The importance of alkenes in biochemistry is well recognized, especially for ethene. Ethene is essential in plant physiology and phenology, functioning as a plant hormone that regulates a myriad of plant processes, including seed germination, root initiation, root hair development, flower development, sex determination, fruit ripening, senescence, and response to biotic and abiotic stresses (Yang and Hoffman, 1984;Reid and Wu, 1992;Lin et al., 2009). All plants and all plant parts produce ethene (typically called ethylene in the plant biology literature), a discovery first made in the 1930s from ripe apples (Gane, 1934). Consequently, ethene is widely used as a ripening agent for plants and plays an important role in the storage and preparation of agricultural commodities. As a plant hormone that responds to various stresses, the ethene source is likely to respond to land and climate modifications. Because of its agricultural importance, the biochemistry of ethene has been well studied by plant physiologists, while the biochemistry of the other light alkenes, such as propene and butene, remains unknown.

Global biogenic volatile organic compound (BVOC) emissions have been simulated for the year 2000 using the MEGAN 2.1 algorithms in the land surface component, CLM4, of the Community Earth System Model CESM (Guenther et al., 2012). Isoprene alone accounted for roughly half of the total annual BVOC emissions by mass at ~535 Tg yr$^{-1}$. The light alkenes, in contrast, only accounted for 5 % of the total emissions. However, the algorithms for light alkene emissions



are based on the very limited field and laboratory measurements described above, meaning that the potential for light alkenes may be much greater than this, especially for ecosystems whose BVOC emissions are not isoprene dominated.

The present study seeks to: a) describe the development and deployment of a continuous REA system to measure net ecosystem fluxes of light hydrocarbons at hourly intervals; b) provide the first net ecosystem flux measurements of light
alkenes from a ponderosa pine forest during the growing season; c) place these results in the context of OH reactivity of other BVOCs that were measured at the site previously; and d) develop emissions parameterizations based on environmental factors for entry into the MEGAN model.

## 2. Site description

In the summer of 2014, a field campaign was conducted at Manitou Experimental Forest Observatory (MEFO) in
the Front Range of the central Rocky Mountains (39.1 °N, 105.1 °W, 2280 to 2840 m a.s.l.), located roughly 100 km south/southwest of Denver, Colorado, USA (**Fig. 1**). The forest is predominantly ponderosa pine with a median tree age of ~50 years and average canopy height of 18.5 m (Ortega et al., 2014). Other local vegetation includes Douglas fir, aspen, mixed conifer, and an understory of primarily grasses. Soils have low organic matter content (1-4 %) and good drainage (i.e., rapid permeability ~50-150 mm h$^{-1}$); soil depth to bedrock averages 1 to 1.8 meters (Ortega et al., 2014).
The climate at MEFO can be described as cold-moderate and dry (430 mm average annual precipitation). Summers are characterized by low humidity and feature hot days (average highs between 22 and 26 °C) with frequent thunderstorms. Long-term observations indicate that the about half of the annual precipitation falls during the summer (National Weather Service, 2016).

The Manitou Forest research site was initially established by the USDA Forest Service in 1936
(http://www.fs.usda.gov/manitou/). In 2008, the National Center for Atmospheric Research (NCAR) established MEFO as part of the Bio-hydro-atmosphere interactions of Energy, Aerosols, Carbon, H$_2$O, Organics and Nitrogen (BEACHON) project. The infrastructure at the site includes a 28-meter walk up "chemistry tower" with mobile laboratory containers located at the base, with line power and temperature control. As part of the BEACHON project, two major field intensives were conducted: BEACHON-ROCS (Rocky Mountain Organic Carbon Study) in 2010 and BEACHON-RoMBAS (Rocky
Mountain Biogenic Aerosol Study) in 2011. Ortega *et al*. (2014) provide a detailed description of the site as well as an overview of the BEACHON projects between 2008 and 2013.

As a result of the BEACHON projects, meteorological and gas-phase measurements have been made on the chemistry tower for multiple consecutive growing seasons. Since 2009, these measurements have included: wind speed and direction, temperature, humidity and pressure (2D sonic anemometer, Vaisala WXT520), and photosynthetically active
radiation (PAR) at 4 locations from the ground-level to the top of the tower (LiCOR LI190SA and Apogee LQS sensors). Direct and diffuse beam PAR (Delta T instruments BF3) were also measured at the top of the tower, ~28 m above ground level (a.g.l.) (Ortega et al., 2014).





The MEFO site is located in a gently sloping drainage valley, with air draining to the north. At nighttime the mountain to valley flow prevails, with winds largely from south to north. During the daytime, southerly flow also occurs, but there is much more variability in wind direction (Ortega et al., 2014).

In this field campaign, net ecosystem fluxes of light alkenes were measured from June 25 to August 9, 2014 (day of year (doy) 176-221), with a lapse between June 29 at noon to July 16 at noon (doy 180-197), owing to instrument problems. Understory fluxes were measured during a case study day on September 2, 2014, after relocating the equipment to a lower measurement height (2 m a.g.l.). The average temperature and precipitation total during this field campaign was 15.9 °C and 210 mm, respectively. On a monthly scale, June 2014 was dry (16.1 °C, 8 mm), July was notably wet (16.6 °C, 151.3 mm) and August was consistent with long term observations (14 °C, 74 mm). Several notable precipitation events occurred on July 12[th] (doy 193, 25 mm), July 25[th] (doy 206, 14 mm) and July 30[th] (doy 211, 13 mm). A longer lasting precipitation event was recorded during July 15[th]-17[th] (doy 196-198, 30 mm), during which time hail was also observed (e.g., July 16, doy 197).

Over the time scale of this field campaign, the air temperature exhibited three synoptic scale fluctuations, lasting about 2 weeks each. These slow fluctuations coincided with fluctuations in ambient pressure and can be explained by changes in local weather systems. On sunny days, net radiation reached 880 W m$^{-2}$, yielding up to 2000 µmol m$^{-2}$ s$^{-1}$ of photosynthetically active radiation (PAR). Duration of daylight was almost 15 hours per day. Hourly time is reported here as Mountain Standard Time (MST = UTC - 7 hours).

## 3. Methods

To quantify net ecosystem exchange of biogenic hydrocarbons, we employed a relaxed eddy accumulation (REA) sampling system coupled to an automated gas chromatography system with flame ionization detection (GC-FID). The REA sampling system was located near the top of the chemistry tower while the gas measurement systems were located in the laboratory at the base of the tower. The following sections describe the REA theory, the REA instrumentation and setup, the automated GC-FID system, and the additional measurement systems deployed during these experiments.

### 3.1. Relaxed Eddy Accumulation (REA) theory

Net ecosystem fluxes for a suite of hydrocarbons were measured on an hourly basis using the relaxed eddy accumulation (REA) method. REA is a micrometeorological flux measurement technique that permits *in situ* flux measurements for chemical species that cannot be measured at the high frequency required for eddy covariance techniques (Businger and Oncley, 1990). To date, no light alkene sensor meets the requirements for detection limit, accuracy, sensitivity and response time for eddy covariance measurements in natural ecosystems. REA systems have been successfully used for other biogenic volatile organic compounds, including isoprene (Bowling et al., 1998;Guenther et al., 1996;Haapanala et al., 2006) and OVOCs (Schade and Goldstein, 2001;Baker et al., 2001).





The REA technique is described in detail in (Businger and Oncley, 1990); therefore, only a brief description is provided here. Air samples are conditionally sampled into an up-draft reservoir, a down-draft reservoir, or a neutral bypass, controlled by fast response valves that respond to high frequency 3-D sonic anemometer measurements of the vertical wind velocity ($w$). Mean vertical wind velocity ($\overline{w}$) is determined for a flux averaging period, and the instantaneous vertical wind velocity is calculated (w' = w(t) −$\overline{w}$). The REA method is derived from the eddy accumulation method (Desjardins, 1977), but 'relaxes' the requirement of sampling at flow rates proportional to the vertical wind speed. In both methods, a turbulent flux is derived from the differences between averaged concentrations in the up- ($\overline{c^+}$) and downdraft ($\overline{c^-}$) reservoirs collected over some flux averaging period (typically 30-60 min). In the surface layer, the concentration differences are scaled by the standard deviation of w ($\sigma_w$) and the dimensionless Businger-Oncley parameter ($b$) to yield the vertical flux (**Eq. 1**):

$$F = b\ \sigma_w\ (\overline{c^+} - \overline{c^-}) \tag{1}$$

In theoretical solutions, $b$ was found to be a weak function of atmospheric stability (Businger and Oncley, 1990). Wyngaard and Moeng (1992) simulate $b$ to be fairly constant ($b \sim 0.627$) assuming a Gaussian joint probability density function between $w$ and $c$. Empirical approximations based on direct eddy covariance measurements show some variation of the b-coefficient on a diurnal basis, and although it varies for different scalars, estimates usually fall in the range of $0.51 < b < 0.62$ (Katul et al., 1996;Ruppert et al., 2006;Baker, 2000;Pattey et al., 1993;Baker et al., 1992). Consequently, a dynamic $b$ value is often used, calculated for each REA averaging interval based on concurrent eddy covariance (EC) measurements of a proxy scalar under the assumption of scalar similarity (Pattey et al., 1993). In this case, $c$ is replaced with the proxy scalar of temperature, measured by the sonic anemometer. The value of $b$ can be calculated from the sonic temperature and by rearranging equation 1 as follows (**Eq. 2**):

$$b = \frac{(\overline{w'T'})}{\sigma_w(\overline{T^+} - \overline{T^-})} \tag{2}$$

where ($\overline{w'T'}$) is the covariance between instantaneous fluctuations of w and temperature, i.e., the heat flux, averaged over the chosen time interval and ($\overline{T^+}, \overline{T^-}$) are the mean temperatures during up- and downdraft sampling, respectively. Ruppert *et al.* (2006) investigated scalar similarity between water vapor, sonic temperature and carbon dioxide and found a diurnal pattern in scalar correlation coefficients leading to an error of $F_{REA} \leq 10\ \%$.

To increase accuracy of conditional sampling and maximize the signal to noise ratio in $\Delta\overline{c}$, samples during very small $w'$ are discarded via a neutral bypass as part of a "deadband"(Baker, 2000). For each flux averaging interval, a symmetrical threshold ($w_0$) around the mean wind velocity is applied, whereby the updraft reservoir is sampled when $w' \geq w_0$ and the downdraft is sampled when $w' \leq -w_0$. Oncley *et al.* (1993) analytically solved the ratio between an increase in uncertainty of $\overline{c}$, due to shorter sampling intervals with increasing $w_0$, over an improvement in the signal to noise ratio and report an optimum at $w_0 = 0.6\ \sigma_w$, which was used in this study. For each flux averaging interval, the Businger-Oncley



parameter is computed from Eq. 2 using the same deadband. The deadband-related increase in $\Delta\bar{T}$ consequently leads to smaller $b$ values that are $\sim 0.4$.

In REA measurements, both $\bar{w}$ and $\sigma_w$ need to be initialized in real time to determine what constitutes an up- and down-draft within each flux averaging interval. Based on the analysis of Turnipseed *et al.* (2009), we chose to use $\bar{w}$ and $\sigma_w$ from the previous flux averaging interval.

**3.2. REA Instrumentation**

The physical REA instrumentation consists of two subsystems: (1) an air sampling subsystem to segregate the sample flow into an up- and down-line (or neutral bypass line) according to the vertical wind velocity, and (2) a reservoir system, for storage, transfer and evacuation of the sampled air (**Fig. 2**). The subsequent description follows the flow of air through the system.

(1) The air sampling sub-system consisted of a sonic anemometer and segregator box, both mounted 25.1 m a.g.l. on the end of a 1.2 m boom (metal cross beam) extending outward from the top level of the walk-up chemistry tower. Vertical wind velocity was measured with an ultrasonic anemometer (Model 81000, R.M. Young, Traverse City, MI, USA), which transmitted data at 5 Hz frequency via RS-232 to a CR-1000 data logger (Campbell Scientific Inc., Logan, UT, USA). A 75 cm long 1/8" outer diameter by 1/16" inner diameter PTFE tube (EW-06605-27, Cole Parmer, Vernon Hills, IL, USA) was attached to the sonic anemometer (horizontal offset $\approx 0$ cm and vertical offset = 10 cm with respect to the center of the anemometer's measurement path). Sample air was drawn into the segregator box (also mounted on the boom) via a micro-diaphragm pump (UNMP805, KNF Neuberger Inc., Trenton, USA), with airflow restricted by a stainless steel needle valve. The segregator split the airflow into an up-, down- and neutral line by two logger controlled solenoid valves ($V_{up}$ and $V_{dn}$, **Fig. 2**) (MP12-62M, Bio-Chem Fluidics Inc., Boonton, NJ, USA). The neutral line was activated when vertical wind velocities fell into the deadband (see **Sect 3.1** above). Neutral airflow was directed through an airflow sensor (AWM3300V, Honeywell International Inc., Morris Plains, NJ, USA) and finally vented out of the segregator.

(2) The reservoir sub-system was mounted on a platform 1 m below the sonic anemometer to collect updraft and downdraft air into two separate sample containers for temporary storage and subsequent analysis. After passing the segregator, sample air was directed either into an "up" bag or a "down" bag (10 Liter Tedlar® bag 231-10, SKC Inc., Eighty Four, PA, USA), controlled by 3-way lift solenoid valves $V_1$ and $V_2$ (**Fig. 2**). All valves of the reservoir system were identical and connected by 1/8" OD PTFE tubing (EW-01540-17 and EW-06605-27, Cole Parmer, Vernon Hills, IL, USA). There were two sets of up and down bags (set $A_{up}/A_{dn}$ and set $B_{up}/B_{dn}$), allowing one pair of bags to be analyzed while the other set was simultaneously used for sampling (60 min).

For the sample set being measured, air from each bag was transferred sequentially (18 min each) through solenoid valves $V_{4u}$ or $V_{4d}$ (**Fig. 2**). Two samples lines (1/4" PTFE tubing wrapped in foam insulation) extended down to the





laboratory trailer at the base of the tower, and air samples were drawn from the reservoir bags to the gas chromatograph (see next section).

To address the potential issue of different storage time in the bags, the order of sample analysis alternated between each hourly flux sampling interval (e.g., 1 pm: up bag, down bag; 2 pm: down bag, up bag). After the transfer, airflow to the
GC was shut off and the remaining air in the up or down reservoir bag was evacuated for 15 minutes through solenoid valve $V_{3u}$ or $V_{3d}$, respectively, using a vacuum pump (UNMP805, KNF Neuberger Inc., Trenton, NJ USA) (**Fig. 2**), with less than 2 % carry over from one sample to the next Additional details are described in the **Supplementary Information** section.

### 3.3.  REA Processing and Quality Control

Real time measurements of vertical wind velocity (w) were collected on a data logger (CR1000 Campbell Scientific
Inc., Logan, USA), which also relayed the signal following the sampling lag time (see **Supplementary Information**) to control the segregator sampling line valves, $V_{up}$ and $V_{dn}$, accordingly. The high frequency time series of sonic temperature (T) were stored in the data logger's memory for subsequent calculation of the covariance of w and T: $(\overline{w'T'})$. Sonic temperature was also conditionally averaged into $\overline{T^+}$ and $\overline{T^-}$ for calculation of the b-coefficient (**Eq. 2**). At the end of each flux averaging interval, $\overline{w}$ and $\sigma_w$ were calculated by the data logger and used to initialize the deadband for the following
sampling hour as well as to compute the instantaneous fluctuations of vertical wind speeds (w').  In addition, the logger also triggered the bag selection valves ($V_1$ and $V_2$) when switching to the other pair of up- and downdraft reservoirs (set A versus set B bags, **Fig. 2**).  For quality control, the volume of sampled air in each bag, the volume of expelled neutral air and the average sampling flow rate were saved on the data logger's memory.  Quality control for each hourly REA flux measurement was checked against eight potential flags associated with the sample volumes, meteorological conditions or
footprint analysis **(Fig. S1, Supplementary Information).**

Flux detection limits ($F_{min}$) were calculated by (**Eq 3**):

$$F_{min} = b \, \sigma_w \, 2\sigma_{c\_std} \tag{3}$$

where $2\,\sigma_{c\_std}$ is the analytical precision based on two standard deviations of hourly repeated GC-FID runs of the calibration standard (see **Sect. 3.6** below). The lowest flux detection limit was achieved for isoprene ($F_{min}$ = 3.4 μg m$^{-2}$ h$^{-1}$), followed by
ethene and butene  ($F_{min}$ = 4.1 μg m$^{-2}$ h$^{-1}$) and propene ($F_{min}$ = 4.7 μg m$^{-2}$ h$^{-1}$).

### 3.4.  Understory REA fluxes

Understory flux measurements were performed on a single day, September 2, 2014 (about a month following the main experiment), to provide insight on the magnitude of fluxes that may be emanating from the surface instead of the tree canopy.  These understory fluxes were measured by mounting the REA sampling system to a separate smaller scaffold, with
the inlet line and sonic anemometer placed at 2 m a.g.l.  Hourly fluxes were measured starting at 6 a.m. and ending at 5 p.m.,





with the up and down bag samples being transferred to electropolished stainless steel canisters for later analysis in the laboratory on the same gas chromatograph used during the field season.

The challenge with understory measurements is that they are prone to sampling artifacts due to flow distortion and low wind speeds. Furthermore, turbulence tends to be intermittent, and there is a lack of universal theories on sub-canopy
turbulence characteristics, i.e. (co)spectral models (Launiainen et al., 2005).

In this study, the understory turbulence (defined here as the standard deviation of vertical wind), evolved over the course of the day from 0.04 m s$^{-1}$ at night/early morning to over 0.1 m s$^{-1}$ at 9 MST to a maximum of ~0.4 m s$^{-1}$ (**Fig. S2**). In previous sub-canopy flux studies, a $\sigma_w$ mixing criterion was empirically determined at 0.1 m s$^{-1}$ (Launiainen et al., 2005). Thus, measured fluxes in periods with insufficient mixing (small $\sigma_w$) do not represent the real surface-atmosphere exchange.
Our observations support the use of a similar criterion: sensible heat fluxes were highly variable under low turbulence conditions but showed weak dependence on $\sigma_w$ with increasing $\sigma_w$. A site-specific $\sigma_w$ threshold was determined at 0.4 m s$^{-1}$.

### 3.5. Gap filling model

Flux measurement time series are often fragmented due to questionable turbulence statistics, unfavorable wind directions or sensor failure. Hence diurnally or seasonally averaged fluxes can be biased if time series are not gap filled.
Gap filling was performed here using an artificial neural network (ANN) approach (Moffat et al., 2007;Papale et al., 2006). Input variables included air temperature, photosynthetically active radiation, water vapor flux and standard deviation of the vertical wind speed. Prior to gap filling, inputs were normalized and gap-filled with average values from the surrounding days (if available) or interpolated values. Inputs variables (n = 2472, each) were then divided in up to k = 20 clusters via the k-means method, a cluster analysis tool which partitions n observations into k ≤ n clusters by minimizing the inner-cluster
variance. From those clusters explanatory data was proportionally sampled into train, test and validation subsets. This procedure aims at avoiding a bias in network training towards data subsets with better data coverage. In total, 20 extractions out of these subsets were performed and run for 5 network architectures with increasing complexity. The best architecture for each of the 20 extractions was chosen by lowest root-mean-square-error (through comparison with the validation subset, which is not used for training the networks at all) and lowest complexity and then used to compute a predicted flux. Gap
filling was finally performed using the median of the 20 resulting predictions.

Goodness of prediction was quantified by correlation coefficients between median prediction and measured data, as well as by root-mean-square-error (rmse). For ethene: $r^2$ = 0.70 and rmse = 32.1 µg m$^{-2}$ h$^{-1}$; for propene: $r^2$ = 0.71 and rmse = 27.7 µg m$^{-2}$ h$^{-1}$, for butene: $r^2$ = 0.80 and rmse = 8.6 µg m$^{-2}$ h$^{-1}$; and for isoprene: $r^2$ = 0.64 and rmse = 28.6 µg m$^{-2}$ h$^{-1}$.





### 3.6. GC-FID measurement

Hydrocarbons ($C_2$-$C_5$ alkenes including isoprene, $C_2$-$C_6$ alkanes, acetylene and some aromatics) were measured with a gas chromatograph with a flame ionization detector (GC-FID) (**Fig. S3**). The automated GC-FID was developed originally for aircraft operation, with 45 hydrocarbons resolved on the capillary column with a detection limit of 2 to 5 ppt for a 350 cm$^3$ STP sample (Goldan et al., 2000;Kuster et al., 2004). The system was modified here to optimize light hydrocarbon measurements using 20 minute run times, and calibration standards were analyzed in between sample runs to produce daily calibration curves, from which concentrations were derived (**Supplementary Information**). This study focused on ethene, propene, isoprene, acetylene, benzene and the three butene isomers (trans-2-butene, 1-butene and cis-2-butene), which were all well resolved by the chromatography. However, the trio of butene isomers had retention times that were clustered together, and these were all present in equal amounts in the calibration standards. Only one of the butene isomers showed consistently significant signals in this study, and this compound was identified tentatively as cis-2-butene based on its retention time. This compound is reported in this study as 'butene' to account for its molar mass and chemical makeup while allowing for the uncertainty of the specific isomer being measured (**Supplementary Information**).

### 3.7. Eddy Covariance $H_2O$ and $CO_2$ flux measurements

Between 2009 and 2014, turbulent fluxes of $CO_2$, water, heat and energy were measured at MEFO (Ortega et al., 2014) using the eddy covariance (EC) method (Baldocchi et al., 1988). An ultrasonic anemometer (CSAT3, Campbell Scientific, Logan, USA) was mounted at 25.1 m measurement height, along with a weather transmitter (WXT520 Vaisala, Vantaa, Finland) to measure absolute temperature and relative humidity. Air was drawn from the tower through a Teflon inlet line into the trailer and measured for $CO_2$ and water vapor measurements using a closed-path IRGA (Li-7000, LI-COR Biosciences, Lincoln, USA). In this study, fluxes were averaged for 30 minute intervals and underwent a quality control scheme including a test on stationarity and on the integral turbulence statistics (Foken and Wichura, 1996). Fluxes from periods failing both tests were removed from the data set (13 %); data failing only one test were flagged (53 %).

Analysis of the tower's suitability for micrometeorological measurements had been performed previously during the BEACHON campaigns (Kaser et al., 2013a). Flux source regions (i.e., the flux footprint) for this campaign were computed using an analytical model (Hsieh et al., 2000), and the median 90 % flux footprint recovery during unstable (blue) and stable (green) atmospheric conditions was spatially mapped (**Fig. 3**). 90 % flux recovery stretched up to 1400 m (median 670 m) upwind from the tower for unstable atmospheric conditions and 5000 m (median 2200 m) for stable atmospheric conditions. Data from easterly winds were flagged for suspicious footprints due to the presence of a lightly traveled paved highway approximately 500 m away. Further data with 90 % flux recovery exceeding 1.9 km were flagged due to possible source/sink inhomogeneity.



## 4. Results

### 4.1. Alkene Concentrations

Ambient alkene concentrations, calculated as the average of the up and down bag reservoirs for the same hour-long period and reported as the end time, showed large fluctuations over the course of the field campaign (**Fig. 4**). Averaged daily concentrations were the highest for ethene (303 ± 138 ppt), followed by propene (182 ± 84 ppt), isoprene (148 ± 98 ppt), acetylene (86 ± 44), butene (51 ± 30 ppt) and benzene (44 ± 17). Hence, the relative standard deviations were all between 38 % (for benzene) and 66 % (for isoprene) (**Table 1**).

Ethene, propene, butene and isoprene concentrations exhibited clear diurnal cycles; lowest concentrations were observed at nighttime, with a minimum typically occurring between 04 to 07 MST (**Fig. 5, red points**). From 07 MST onwards, concentrations sharply increased and reached maxima at 13 MST for ethene and propene. Butene and isoprene were also elevated during midday, although concentration peaks were not as pronounced. During the afternoon, all of these compounds showed a slow decrease towards the nighttime minima. In contrast, benzene showed only minor enhancement in concentration during the daytime, and acetylene concentrations showed no measurable diurnal cycle (**Figs. 4 and 5**).

Gaps in the measurement period complicate the picture for larger time scale fluctuations in concentrations. The highest concentrations for ethene, propene and butene occurred between days 198 and 206 during midday. Ethene and propene also had high concentrations in the early measurement period between days 176 to 181 when butene concentrations were not monitored. The highest daytime isoprene concentrations occurred between days 200 and 208, also during midday. Acetylene had two periods of higher concentrations, between days 197 to 201 and days 220 to 223, with the highest concentrations occurring either in the daytime or at night. Benzene showed no obvious temporal trends.

### 4.2. Alkene Fluxes

Approximately 450 net fluxes (**Fig. 6**) were quantified over the course of the summer, of which 120 individual fluxes are of high quality (no flags) and a further 159 are of medium quality. Ethene had the highest daytime emissions at $131 \pm 67$ µg m$^{-2}$ h$^{-1}$, followed by propene, isoprene and butene ($109 \pm 61$, $46 \pm 41$ and $43 \pm 22$ µg m$^{-2}$ h$^{-1}$, respectively, **Table 1**). Gap filling REA fluxes (**Fig. 6**) using artificial neural networks (i.e., modeled results) did not significantly change average fluxes for these compounds (**Table 1**), suggesting that the selectivity of quality controlled measurements does not unduly bias diurnal averages. When negative alkene fluxes were measured, they usually failed quality control owing to stable nocturnal atmospheric conditions; however, 1-6 % of the reported QC-ensured fluxes suggest apparent uptake at night with only n = 4 for ethene (-32 µg m$^{-2}$ h$^{-1}$), n = 6 for propene (-22 µg m$^{-2}$ h$^{-1}$), n = 2 for butene (-10 µg m$^{-2}$ h$^{-1}$), and n = 17 for isoprene (-18 µg m$^{-2}$ h$^{-1}$) being larger than flux detection limits.

The time series of alkene fluxes show distinct diurnal patterns of emissions which are similar for ethene, propene and butene (**Fig. 5, blue points**). Diurnally-averaged fluxes show low (but generally positive) nighttime emissions, a rapid



rise during the morning, elevated fluxes throughout the day from 10 to 17 MST, and decreasing fluxes in the evening. Isoprene fluxes on average showed a similar pattern, but decreased earlier in the afternoon (15 MST) and had roughly zero flux at nighttime. In contrast, acetylene and benzene showed no diurnal flux patterns and scatter around zero: $1 \pm 13$ µg m$^{-2}$ h$^{-1}$ for acetylene and $-2 \pm 17$ µg m$^{-2}$ h$^{-1}$ for benzene (**Fig. 5, Table 1**).

In addition to the diurnal patterns, multi-day (~5 day) fluctuations were visible in the measured peak daytime fluxes for the alkenes (**Fig. 6**). Daytime maximum emissions rose and fell 50 % between days 198-205 and again between days 215-220. The pattern resembles the broad temporal trends in temperature, radiation, and water flux (**Fig. 6**).

**Table 1.** Average light alkene concentrations and fluxes at Manitou Forest between June 24-August 9, 2014; understory fluxes on

September 2, 2014; and flux detection limits.

| | Compound | Average Concentration ppt | Measured Flux µg m$^{-2}$ h$^{-1}$ | Modeled Flux[a] µg m$^{-2}$ h$^{-1}$ | Daytime flux[b] (measured) µg m$^{-2}$ h$^{-1}$ | Flux under-story µg m$^{-2}$ h$^{-1}$ | Detection limit µg m$^{-2}$ h$^{-1}$ |
|---|---|---|---|---|---|---|---|
| $C_2H_4$ | ethene | 303.2 ± 137.7 | 71.8 ± 65.4 | 69.5 ± 56.3 | 130.9 ± 67.0 | -26.0 ± 27.1 | 4.1 |
| $C_3H_6$ | propene | 181.6 ± 84.0 | 65.5 ± 66.2 | 57.5 ± 50.9 | 109.4 ± 61.3 | -32.8 ± 47.5 | 4.7 |
| $C_4H_8$ | butene | 50.6 ± 29.5 | 19.6 ± 21.6 | 20.3 ± 19.7 | 42.7 ± 21.5 | -12.4 ± 9.4 | 4.1 |
| $C_5H_8$ | isoprene | 147.5 ± 97.8 | 28.1 ± 37.8 | 28.5 ± 35.5 | 62.3 ± 48.3 | 116.1 ± 108.0 | 3.4 |
| $C_2H_2$ | acetylene | 85.7 ± 44.4 | 0.6 ± 12.5 | n/a | 1.1 ± 18.3 | 0.8 ± 7.8 | 13.6 |
| $C_6H_6$ | benzene | 43.8 ± 16.7 | -2.3 ± 16.7 | n/a | -5.5 ± 22.6 | -3.1 ± 7.2 | 5.4 |

[a] Gap filled using artificial neural networks (ANN)

[b] 10-18 MST

### 4.3. Eddy Covariance: $CO_2$, $H_2O$ and energy fluxes

     Over the sampling period (June 24-August 9, 2014), Manitou Forest acted as a net $CO_2$ source of 2.6 g m$^{-2}$ d$^{-1}$ on average (**Fig. 6**). Characteristic diurnal flux patterns show nighttime to morning respiration (2-8 µmol m$^{-2}$ s$^{-1}$) and net $CO_2$

uptake (up to -8.6 µmol m$^{-2}$ s$^{-1}$) between 09 and 18 MST. A simple one-level storage term evaluation was performed (Rannik et al., 2009). The venting of stored $CO_2$ was on the order of magnitude of measured EC fluxes in the morning (06-08 MST), leading to apparent emission during the onset of turbulence. Storage occurred at night (19-24 MST), offsetting for additional ~25 % of measured nighttime respiration. Over the course of a day, the positive and negative storage terms cancel each other out.

The diurnal $CO_2$ flux cycle increased in amplitude following the onset of significant seasonal rainfall. In the first half of the measurement period, June 24 through July 11$^{th}$ (doy 175 through 192), daily maximum and minimum $CO_2$ fluxes





were relatively small, averaging $4.4 \pm 1.4$ and $-3.1 \pm 1.7$ µmol m$^{-2}$ s$^{-1}$, respectively. Following a strong rain event on July 12 (doy 193, between 15 to 17 MST), these averaged $7.5 \pm 2.4$ and $-5.8 \pm 1.6$, respectively, through the end of the campaign on August 9$^{th}$ (doy 193 to 221). During this latter time period, numerous significant rainfall events also occurred (**Fig. 6**).

H$_2$O fluxes have a characteristic diurnal pattern, with negligible fluxes during nighttime, a sharp increase during sunrise (7 MST), maxima at 12 MST, and a steady decrease during afternoon. On overcast days, peak emissions were on the order of 1.2 mmol m$^{-2}$ s$^{-1}$, whereas on sunny days fluxes reached up to 7.8 mmol m$^{-2}$ s$^{-1}$. H$_2$O storage was found to be negligible. As with CO$_2$, the amplitude of water vapor fluxes increased from July 12 (doy 193) onwards. Average daily maximum water vapor fluxes were $2.9 \pm 0.2$ and $5.4 \pm 0.7$ mmol m$^{-2}$ s$^{-1}$ for the measurement periods before and after July 12$^{th}$, respectively.

Sensible heat fluxes (H$_S$) ranged from -100 to 500 W m$^{-2}$. Typical diurnal patterns indicated nighttime inversions from 20-07 MST and peak emissions at 12 MST. Computing the Bowen Ratio (B = sensible heat divided by latent heat fluxes) gives insight into the eco-system's response to water availability. In the dry period prior to day 193, B was strictly > 1 (median = 2), typical for semi-arid water-limited ecosystems. During this time, evaporation was restricted, favoring elevated sensible heat flux. After rainfall events B dropped below 1 (median = 0.4), due to higher latent heat fluxes and hence less sensible heat flux.

### 4.4. Correlations

Above canopy concentrations of ethene, propene and butene were highly correlated ($0.5 < r^2 \leq 0.8$), whereas correlations including isoprene were weaker ($0.2 \leq r^2 \leq 0.28$) (**Fig. 7**). Average molar concentration ratios were: propene/ethene (0.54), butene/ethene (0.17) and isoprene/ethene (0.47).

Similar to the concentration correlations, the net fluxes of ethene, propene and butene show high correlation coefficients with each other $0.4 < r^2 \leq 0.9$, whereas correlations with isoprene are weak $r^2 < 0.2$. Mass flux ratios were: propene/ethene (0.80), butene/ethene (0.28) and isoprene/ethene (0.24).

### 4.5. Understory fluxes

The understory flux measurements on September 2, 2014 can help partition the above-canopy fluxes between surface and canopy sources. Of the eight REA flux samples collected that day, six flux samples exceeded the $\sigma_w$ threshold of 0.4 m s$^{-1}$; the two samples that fell beneath the threshold occurred during the early morning hours. For the light alkenes, the understory fluxes greatly contrasted the above canopy fluxes. The understory REA measurements showed detectable consumption overall for ethene, propene and butene as opposed to the large emissions observed from the above-canopy fluxes (**Table 1**).

In contrast, the isoprene, acetylene and benzene fluxes were in similar ranges to the above canopy fluxes. Isoprene showed relatively large emissions during the day at the surface, which are in the upper range of observed daytime emissions





from the above canopy measurements. Acetylene and benzene showed small fluxes that scattered around zero, similar to the above canopy measurements.

## 5. Discussion

The magnitude and temporal pattern of these light alkene emissions reveal several aspects of trace gas biogeochemistry and atmospheric chemistry from this ecosystem. First, the origin of the light alkenes is deduced to be local and biogenic through an analysis of flux footprint combined with a comparative analysis with other VOCs measured at the site. Second, the results can be put in the context of the prior BEACHON campaigns to demonstrate the relative importance of light alkenes in the overall emission of reactive VOCs from this ponderosa pine ecosystem. Third, the Manitou Forest results can be compared with the few literature measurements of light alkene fluxes in other ecosystems. Fourth, modeled fluxes can be compared to the light and temperature responses for other BVOCS. Finally, the results provide insights regarding the modeling capabilities of global vegetation BVOC models.

### 5.1. The origin of the light alkenes

While isoprene is well known to be a biogenic volatile organic compound, the biogenic sources for the light alkenes are not as well determined. In this study, ethene, propene and butene appear to originate from local sources that are also biogenic in origin, and in particular, from the forest canopy.

The large diurnal fluctuations of both ambient concentrations and net fluxes of the alkenes follow sunlight and temperature cycles, typical for biogenic VOCs. For example, prior studies at Manitou Forest showed that summertime VOCs with diurnal cycles were predominantly biogenic, with highest contributions from 2-methyl-3-buten-2-ol (232-MBO or MBO), methanol, ethanol, acetone, isoprene and to lesser extent, monoterpenes (mostly α-pinene, β-pinene and Δ-3-carene) (Kim. et al., 2010;Greenberg et al., 2012). Diurnal patterns of alkene concentrations agree with observations of the sum of MBO+isoprene and are inverted to the ones of the sum of monoterpenes, where emissions occur throughout day and night and concentrations build up in the shallower boundary layer over nighttime and deplete during daytime sunlight, both due to dilution in the growing boundary layer and due to reactions with $O_3$ and OH (Kaser et al., 2013b).

In contrast, no such diurnal patterns in concentration are observed for the primarily anthropogenic compounds (acetylene and benzene), and their fluxes are near zero (**Table 1**). Acetylene is considered to be a tracer of combustion originating from biomass burning or urban areas (Xiao et al., 2007). The two general periods of elevated ambient acetylene concentrations, between days 197 to 201 and days 220 to 223, did not correspond to the highest concentrations of the light alkenes. Also, elevated acetylene concentrations typically occurred at nighttime, not at midday like the biogenic VOCs. Benzene appeared to have a slight diurnal fluctuation, but this compound may also have a minor biogenic source in addition to its anthropogenic sources (Misztal et al., 2015). In prior studies at Manitou Forest, it was shown that on days with long-





range transport from the front range cities (Colorado Springs, Denver), anthropogenic VOCs were present, although typically at low concentrations, and no significant local anthropogenic emissions were detected in the area around the site (Ortega et al., 2014).

The REA method requires a measurable concentration difference based on vertical winds. Thus, the observation of alkene emissions points to a local source, and the flux footprint during the daytime is predominantly ponderosa pine forest. The vertical concentration gradient of any source outside of the flux footprint would be erased because of mixing by the time it reached the tower, perhaps generating elevated concentrations but no measurable flux. The benzene and acetylene measurements support this; elevated concentrations in ambient air were occasionally observed for these compounds, presumably from distant anthropogenic sources, but they were not associated with emissive fluxes at the site.

The understory measurements demonstrate that these light alkenes are emitted from the forest canopy, not from the surface litter or soils (**Table 2**). In fact, light alkenes showed a small downward flux to the surface, suggesting potential consumption. Very small emission rates of light alkenes from a boreal forest floor in Finland (< 1.8 µg m$^{-2}$ hr$^{-1}$ for ethene, < 0.5 µg m$^{-2}$ hr$^{-1}$ for propene and < 0.05 µg m$^{-2}$ hr$^{-1}$ for cis-2-butene) (Hellén et al., 2006) may also be consistent with the present study, given that the light alkene emissions appear to be from the canopy, not from the forest floor.

In contrast to the light alkenes, surface isoprene emissions were relatively large, comparable in magnitude to the above-canopy emissions during the growing season. The understory included grasses and herbaceous flower plants (forbs), which were not predicted to be significant sources of isoprene. Leaf and needle litter emissions of BVOCs were measured from Manitou Forest previously, and a compound with the ion m/z=69 (such as isoprene) was measured on the PTR-MS. This compound was tentatively identified as pentanal because of the lack of known isoprene-emitting vegetation at the site 20 (Greenberg et al., 2012), but our measurements suggest that a small local isoprene surface source exists. The relatively small fluxes of isoprene are consistent with BEACHON campaign measurements which showed that isoprene amounted to ~10-20 % of MBO concentrations at Manitou Forest (Karl et al., 2014). Benzene and acetylene show negligible fluxes in the understory, similar to above-canopy fluxes. Taken together, these observations suggest that the canopy is the source for the light alkenes and the understory is a source for isoprene.

A direct comparison between tower based and understory fluxes cannot be made because only one REA system was available. However, the light (1300 - 1700 µmol m$^{-2}$ s$^{-1}$) and temperature (20 - 26 °C) conditions during the understory measurements on Sept 2$^{nd}$, 2014, can be inserted into the temperature and PAR parameterizations from the tower-based measurements to calculate expected fluxes (**section 5.4 below**). Doing this yields a predicted isoprene emission of 75 ± 13 µg m$^{-2}$ h$^{-1}$, which is two-thirds of the averaged measured undercanopy flux (**Table 1**) and supports the hypothesis that the 30 understory is the dominant source for isoprene.



### 5.2. In the context of prior BEACHON campaigns

We can assess the relative importance of light alkenes in the overall emission of reactive VOCs from this ponderosa pine ecosystem by comparing the light alkene emissions measured in this study with the other BVOCs measured during the BEACHON campaigns. In order to do this, it is important to place 2014 in the context of prior years using ecological

parameters measured across all of these years. Eddy covariance flux measurements of $CO_2$ and heat allow this type of comparison: $CO_2$ fluxes, PAR and net radiation flux observed from June-August 2014 (**Fig. 6**) were similar to observations made during the 2008-2013 BEACHON campaigns, both in magnitude and seasonal pattern (Ortega et al., 2014). For example, the summer net ecosystem exchange (NEE) is usually positive, while the spring NEE is negative. Also, the increase of $CO_2$ emissions following the onset of precipitation has been observed at this site in previous years. This has been

attributed to the "Birch effect" found in semi-arid, Mediterranean and African ecosystems, whereby precipitation triggers a burst of organic matter decomposition with subsequent $CO_2$ emissions, significantly reducing/inverting NEE in forest ecosystems (Jarvis et al., 2007).

The overall seasonally averaged sum of ethene, propene and butene fluxes is ~150 $\mu$g m$^{-2}$ h$^{-1}$ over the course of the day, and this amount is substantial even in comparison to the other BVOCs previously measured at the site. For example,

the daytime average (10-18 MST) flux of combined light alkenes was ~280 $\mu$g m$^{-2}$ h$^{-1}$. This is approximately 15 % of combined MBO + isoprene flux of 1.84 mg m$^{-2}$ h$^{-1}$ (combined because the PTR-MS measurements were not able to fully discriminate between these compounds), and it is two-thirds of methanol emissions (0.42 mg m$^{-2}$ h$^{-1}$) (Kaser et al., 2013a). Thus, the light alkenes contribute a significant amount of reactive carbon to the atmosphere at this coniferous forest ecosystem and may even play a bigger role in ecosystems that do not emit MBO.

To assess the relative importance of the light alkenes and isoprene to the total OH reactivity of the BVOCs, we utilized the daytime fluxes from this study compared with the MBO, methanol, monoterpenes, acetic acid, glycoaldehyde, acetaldehyde, ethanol, acetone, propanol and formic acid fluxes reported previously for this site (DiGangi et al., 2011;Kaser et al., 2013a). Multiplying mixing ratios of these compounds by their OH rate constants provides a measure of relative OH reactivities (Ryerson et al., 2003;Fantechi et al., 1998;Ravishankara and Davis, 1978;Atkinson et al., 1986;Atkinson et al.,

1997;Baulch et al., 1994;Huang et al., 2009;Picquet et al., 1998). We utilized fluxes instead of concentrations to provide a measure of OH reactivity that is independent of elevated concentrations associated with pollution events and more representative of site specific sources. Accordingly, the dominant BVOC for OH reactivity is MBO, accounting for 62 %, followed by monoterpenes at 11 %, isoprene at 7 %, and acetaldehyde at 6 % (**Fig. 8**). Ethene, propene and butene accounted for 3 %, 5 %, and 3 % of the OH reactivity, respectively. Combined, the light alkenes accounted for 11.5 % of the

total OH reactivity, comparable to the monoterpenes and second only to MBO. Thus, the light alkenes are an important component of the atmospheric chemistry of ponderosa pine forests. It is possible that unmeasured or underestimated emissions of the light alkenes can contribute to the problem of missing OH reactivity observed in other forests, as the





reactive source for the missing OH has the temperature response characteristics of a BVOC (Di Carlo et al., 2004;Mogensen et al., 2011;Nölscher et al., 2013).

### 5.3. Literature comparison of fluxes

Net ecosystem fluxes of light alkenes have been reported for one other forested site: a temperate deciduous forest in Massachusetts (Harvard Forest 42° N, 72° W) (Goldstein et al., 1996). Using a flux gradient method, average emission fluxes were derived for ethene, propene, and butene (1-butene) of 44.1, 28.4, and 13.8 µg m$^{-2}$ h$^{-1}$, calculated as the integrated mean diurnal fluxes between June 1 to October 31, 1993. In the present study, Manitou Forest emissions were larger by factors of 1.5 to 2 (69.5, 57.5, and 20.3 µg m$^{-2}$ h$^{-1}$), respectively. However, this study focused on the summer months of July-August, 2014, and the much higher fluxes are partly a consequence of averaging fluxes over a period of higher temperature and/or PAR. A simple extrapolation for the whole season at Manitou Forest, assuming linear increases and decreases from/to zero during the shoulder months, still yields 20 to 60 % larger seasonal fluxes, suggesting that the coniferous Manitou Forest indeed emits more per unit area than the deciduous Harvard Forest.

In both studies, the fluxes of these alkenes were correlated with each other, although with slightly different ratios. Goldstein *et al.* (1996) report molar ratios of emission of ethene and butene versus propene of 1.8 ± 0.22 (std error) and 0.41 ± 0.06, respectively, whereas this study yielded ratios of 1.1 ± 0.17 and 0.52 ± 0.14 (std error), respectively. While the butene/propene ratio appears to be similar, a key difference is that the butene isomer identified by Goldstein *et al.* (1996), was 1-butene, whereas in this study, the butene isomer is tentatively identified as cis-2-butene.

Strong diurnal cycles of ethene, propene and butene fluxes were observed in both forests, but MEFO fluxes more closely tracked temperature than incident light whereas Harvard Forest was vice versa. This was illustrated both by the temporal synchronicity as well as the stronger correlation between the alkene fluxes and ambient temperature (for MEFO) or PAR (for Harvard Forest) (see **Section 5.4**). At MEFO, ambient temperature usually peaked 1-2 hours after PAR starts declining, similar to the alkene fluxes.

A brief comparison can be made with other observed biogenic emissions of light alkenes. Ethene emission rates from plant shoots compiled by Sawada and Totsuka (1986) averaged 1.5 ng ethene per gram fresh weight (gfrw) per hour, with a range of 0.6-3.2 ng (gfrw)$^{-1}$ h$^{-1}$. Emission rates were combined with biomass and surface area estimates of biomes to derive a net areal flux from coniferous forests for the growing season of 29.8 µg m$^{-2}$ h$^{-1}$ from plant shoots/leaves. This is roughly 40 % of the average ethene flux estimated here (69.5). Given the fact that the prior study was based largely on a very limited number of laboratory incubations of non-arboreal species, it is remarkable that the emission rates are within a factor of three of each other. On the other hand, the emission rates from coniferous forests during the warmest part of the summer appear to exceed the previously assumed upper range of emissions.





### 5.4. Light and Temperature Responses

There is a striking similarity in the multi-day patterns observed in both the biogeochemical fluxes and environmental parameters at MEFO. The meso-scale temporal patterns in the fluxes are illustrated by a rise and fall of peak midday values (**Fig. 6**), such as the one occurring between d.o.y. 198 and 212 followed by another between d.o.y. 202 and 223. A similar pattern is evident in the peak midday $H_2O$ flux, the maximum daily air temperature and the net radiation/PAR. These trends were measured independently, with separate instruments using different methods. The relationship between the fluxes and environmental parameters suggests that sunlight and temperature control the variability in the alkene fluxes and evapotranspiration rates.

To describe temperature and light responses, alkene fluxes have been averaged into bins of (a) 2 °C temperature and (b) 200 µmol m$^{-2}$ s$^{-1}$ PAR classes (**Fig. 9**). A temperature response (F(T)) was parameterized according to **Eq. 4** (Schade and Goldstein, 2001):

$$F(T) = \ \alpha \exp (\beta\, T) \tag{4}$$

where $\alpha$ and $\beta$ are empirical coefficients (**Table 2**) and T is the ambient temperature (°C).

This relationship is similar to ones governing monoterpene and sesquiterpene emissions, which are a function of temperature only. Temperature responses in the 0 < T < 30 °C range follow an exponential function, are fairly similar between individual alkenes and agree well with observations from Schade and Goldstein (2001) for MBO and Kaser *et al.* (2013b) for methanol (**Fig. 9**).

Correspondingly, a light response (F(PAR)) was parameterized according to **Eq. 5** (Harley et al., 1998):

$$F(PAR) = \frac{\alpha\, C_{L1}\, PAR}{\sqrt{1 + \alpha^2 PAR^2}} \tag{5}$$

where $\alpha$ and $C_{L1}$ are empirical coefficients (**Table 2**) and PAR is the photosynthetically active radiation (µmol m$^{-2}$ s$^{-1}$). This relationship was originally developed for emissions of isoprene, whose production is light-dependent.

Temperature responses curves showed higher correlation coefficients than light response curves. Ethene, propene and butene responses were similar to each other and agree remarkably well with the MBO and methanol response parameterization from Harley *et al.* (1998), Schade and Goldstein (2001) and Kaser *et al.* (2013a) (**Fig. 9**). These compounds are also believed to be biogenic in origin and emitted from the canopy during photosynthesis. MBO flux profile measurements show that MBO emissions are light dependent and increase with height up to 12 m (Karl et al., 2014;Ortega et al., 2014). Responses show an almost linear increase at PAR < 1000 and asymptotic behavior at PAR ≈ 2000 µmol m$^{-2}$ s$^{-1}$. The isoprene light response, on the other hand, differed from other compounds, with less of an asymptote at high PAR. This is consistent with respect to the observed source of isoprene in the understory, which is partially shaded. Subcanopy PAR measurements (2 m a.g.l.) during this field campaign were on average a factor of 3.4 smaller than tower-based measurements above the forest canopy.



**Table 2:** Fitted coefficients for temperature and light responses.

| Compound | $F_{30}$ [temp] ($\mu g\ m^{-2}\ h^{-1}$) | $\alpha$ | $\beta$ | $r^2$ | $F_{1000}$ [light] ($\mu g\ m^{-2}\ h^{-1}$) | $\alpha$ | $C_{L1}$ | $r^2$ |
|---|---|---|---|---|---|---|---|---|
| Ethene | 302.9 | 0.0325 | 0.1142 | 0.94 | 130.0 | 0.001716 | 1.1577 | 0.88 |
| Propene | 285.9 | 0.0226 | 0.1263 | 0.95 | 110.7 | 0.001523 | 1.1969 | 0.83 |
| Butene | 96 | 0.0237 | 0.1247 | 0.93 | 37.7 | 0.001263 | 1.2769 | 0.86 |
| Isoprene | 185.3 | 0.0095 | 0.1554 | 0.94 | 42.7 | 0.000681 | 1.7974 | 0.80 |

### 5.5. Parameterization of fluxes for modeling

The light alkenes (ethene and propene) are included in the Model of Emissions of Gases and Aerosols from Nature version 2.1 (MEGAN 2.1), which is used to determine the BVOC input into the atmosphere from terrestrial and oceanic ecosystems. Perhaps the best characterized BVOC in MEGAN 2.1 is isoprene, and it is noteworthy that the modeled parameters for isoprene flux in this study are in excellent agreement with MEGAN 2.1., with nearly identical parameterizations (CL1 = 0.0007 $\alpha$ = 1.73 in this study, CL1 = 0.0007 $\alpha$ = 1.74 in MEGAN 2.1).

In MEGAN 2.1, ethene is classified as a 'stress VOC' owing to its known biochemical production during times of abiotic and biological stress (Abeles et al., 2012), while propene and butene are classified as 'other VOCs'. In this study, propene and butene fluxes highly correlate with ethene fluxes and show a very similar light and temperature response. Hence, our results suggest that propene and butene can be categorized together with ethene. In MEGAN 2.1, global butene emissions are only 30 % of ethene and 50 % of propene, which is similar to the ratios found here (30 % and 40 % respectively). Modifying the light and temperature parameterizations for light alkenes in the vegetation emissions model will lead to a corresponding increase in estimated global emissions for these compounds. This would generally support the conclusion of Goldstein *et al.* (1996) that "terrestrial biogenic emissions could provide a significant global source for two important reactive olefins, propene and 1-butene", with the caveats that the specific butene isomer remains in question and that other terrestrial ecosystems need to be surveyed.

### 6. Conclusions

The Relaxed Eddy Accumulation technique coupled with GC-FID analysis proved to be suitable to quantify fluxes of ethene, propene, butene and isoprene from a coniferous forest canopy. This study demonstrated that coniferous forests can be significant sources of these compounds, and that the emissions of the light alkenes alone can constitute roughly 15 %





of the dominant emission flux of 2-methyl-3-buten-2-ol (MBO) and roughly two-thirds of methanol fluxes. The three light alkenes (ethene, propene and butene) can constitute roughly 10 % of the overall OH reactivity associated with BVOCs. Thus, the emissions of light alkenes should be included in the overall emissions of reactive organic compounds in the forest atmosphere. Presently, little is known about flux magnitudes of light alkenes in different ecosystems, e.g., broadleaf

evergreen forests of the tropics. In ecosystems not dominated by MBO or isoprene, light alkenes may be major components of the overall BVOC emissions. At Manitou Forest, ethene, propene and butene are light and temperature driven and appear to originate from within the canopy. Hence the light alkenes may scale with the photosynthetic activity of the ponderosa pine trees. While isoprene emissions are also light and temperature dependent, this compound appears to emanate from near surface vegetation, not the canopy. Due to their reactivity with the hydroxyl, ozone and the nitrate radical, we suggest that

these compounds be incorporated in future BVOC-atmospheric chemistry modeling studies. More enclosure-based and ecosystem-scale measurements in forested ecosystems (especially in boreal and tropical forests) are needed to assess plant functional type specific emission inventories.

### 7. Author contribution

R.C.R. planned and oversaw the MEFO 2014 field campaign; M.J.D. analyzed a majority of the data, applied the flux

models and produced the graphs; A.T. designed, built, deployed and verified the REA system; C.W. upgraded the GC-FID system and provided instrumental support; J.O. and J.S. managed MEFO instrumentation and logistics with NCAR; S.S. and L.M. conducted the field work; A.K., J.G. and B.L helped with GC measurements and calibrations; A.G. and J.d.G. were co-PIs, supporting laboratory and field work and intellectual direction of the project. R.C.R. prepared the manuscript with contributions from all co-authors. A.G. is an editor for A.C.P.; all other authors declare that they have no conflict of

interest.

### 8. Acknowledgements

We thank the USDA Forest Service and S. Alton for access, facilities and support at MEFO; S. Shertz and S. Gabbard for research support at NCAR; A. Cowart for cartography support (Fig. 1); A. Goldstein for valuable advice and feedback; and B. Miller and W. Kuster for GC support. RCR thanks CIRES/NOAA and NCAR for their visiting fellows programs. L.

Martinez thanks the NOAA Hollings undergraduate scholarship program. This research project was supported primarily by NSF Atmospheric Chemistry. MEFO is supported by NCAR, and NCAR is supported by the NSF.



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





**Figures**

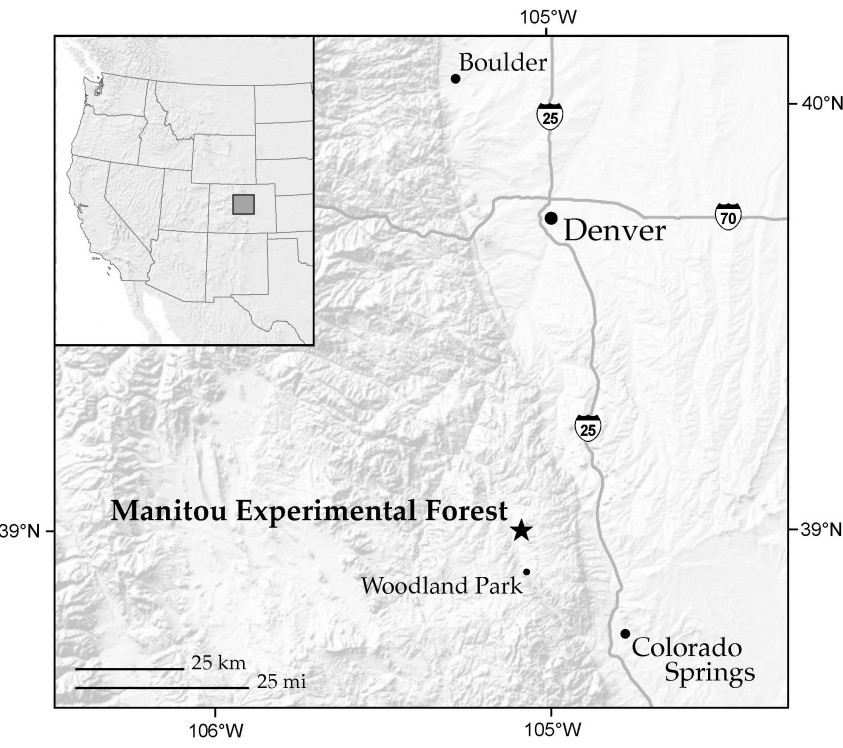

**Figure 1.** The Manitou Experimental Forest Observatory, located in the Front Range of the Rocky Mountains, is shown relative to the cities of Denver, Boulder, Colorado Springs and Woodland Park in Colorado. Interstate highways 25 and 70 are shown.





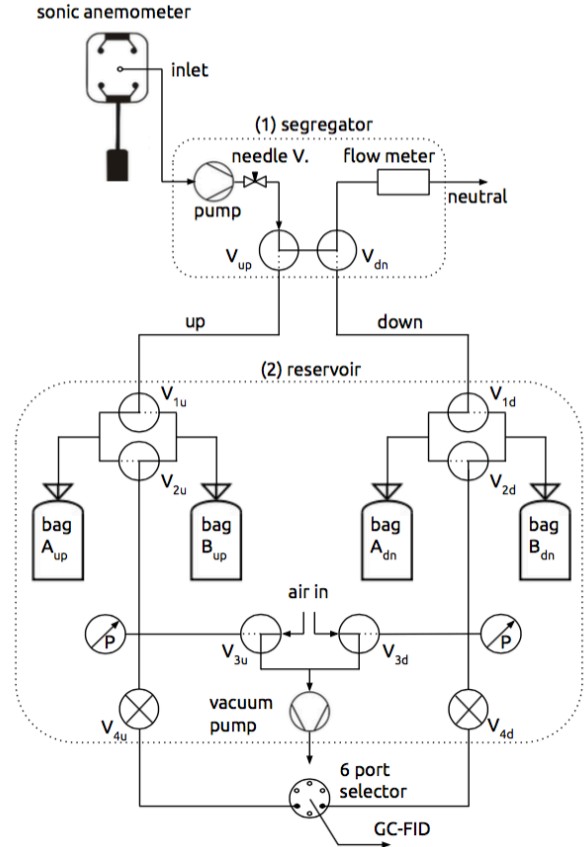

**Figure 2.** The Relaxed Eddy Accumulation (REA) system is comprised of: (1) a segregator subsystem and (2) a reservoir subsystem. Sample valves indicated by V, with updraft (up) and downdraft (dn) air sampling valves and bag reservoirs shown.



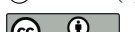

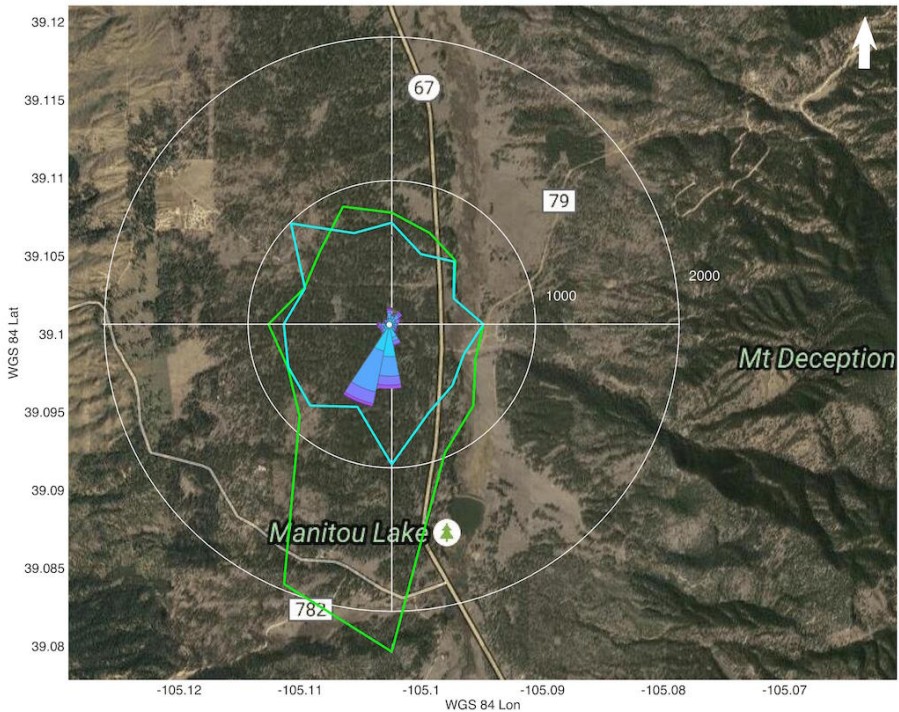

**Figure 3.** Aerial image of the tower site and the flux footprint (median 90 % recovery) during unstable (blue) and stable (green) atmospheric conditions in this field campaign. Background imagery from Google Earth.





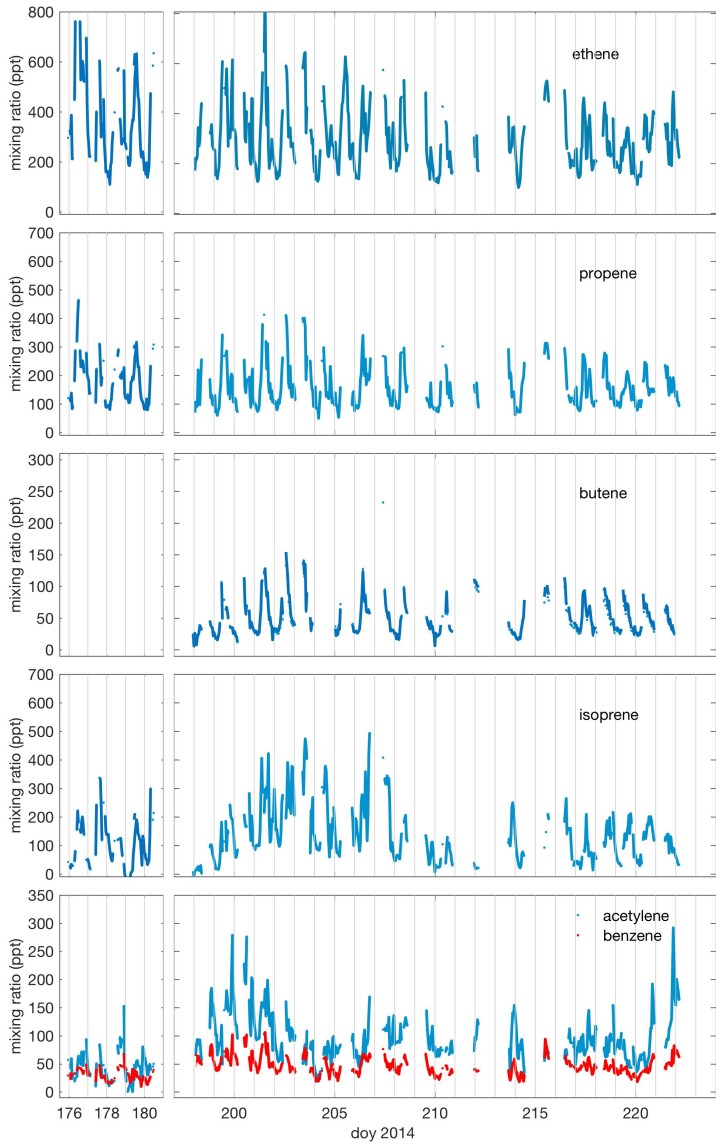

**Figure 4.** Hourly averaged ambient concentrations of alkenes, acetylene and benzene at Manitou Forest. Periods of missing data due to instrumental maintenance or incomplete chromatography.




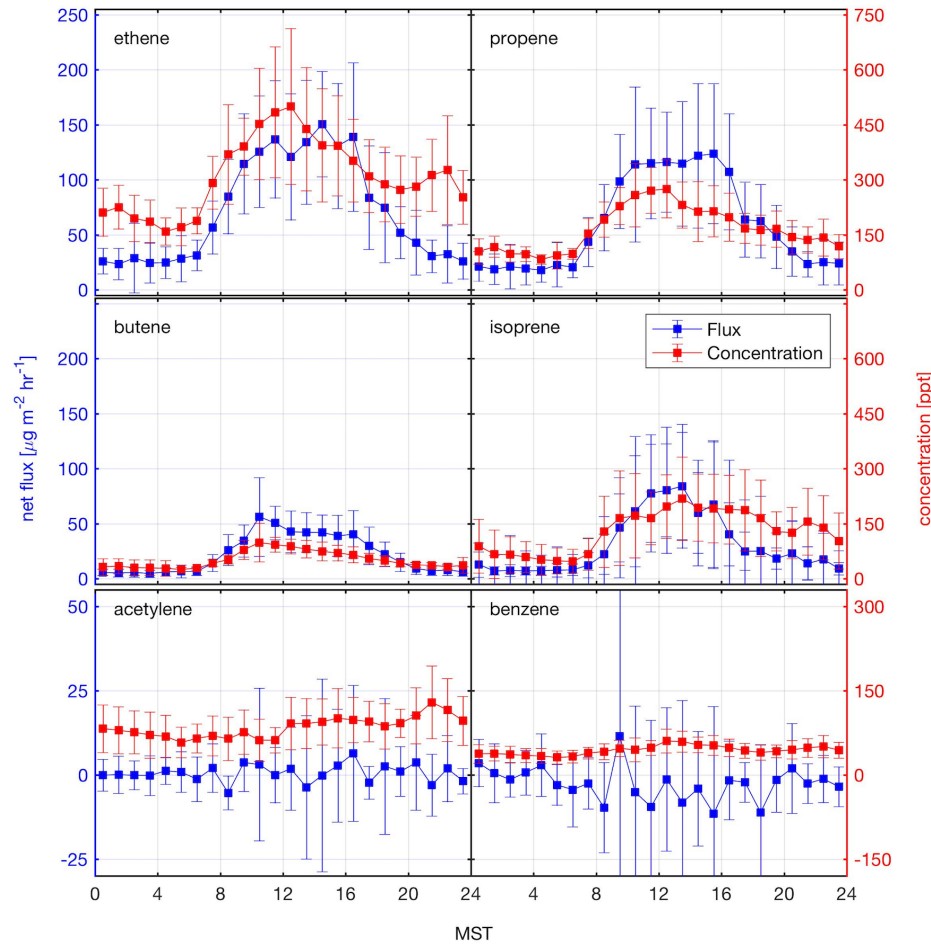

**Figure 5.** Averaged diurnal patterns of alkene, acetylene and benzene concentrations (red) and their fluxes (blue) with error bars indicating ±1 σ.





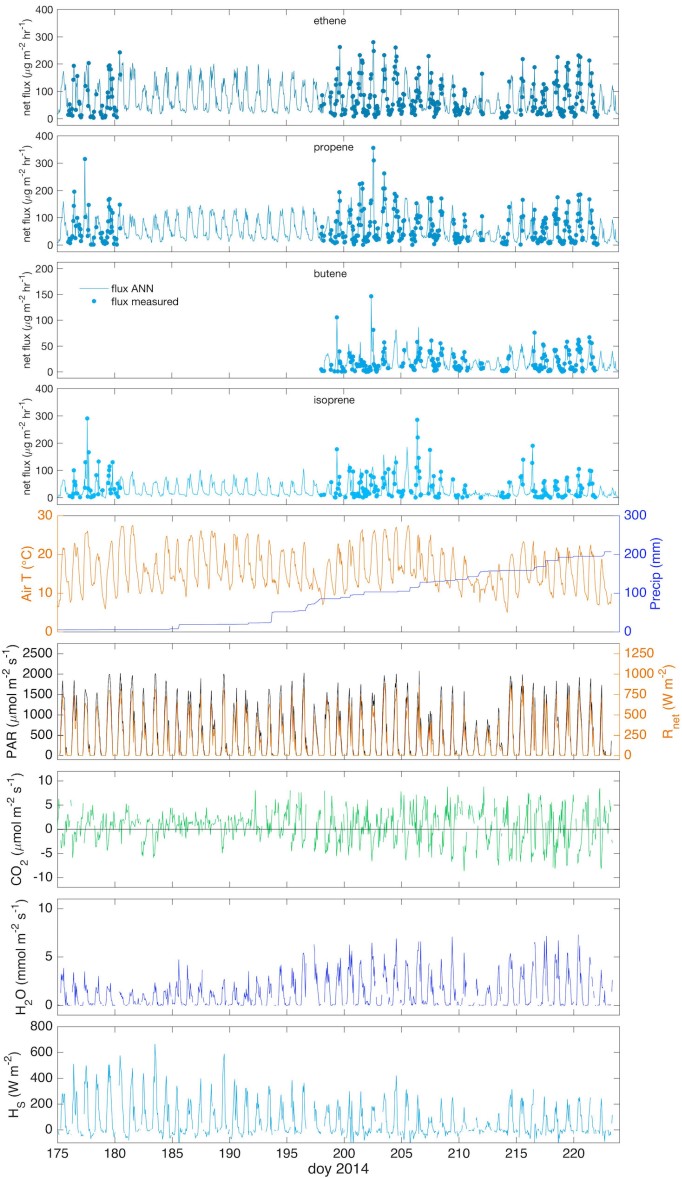

**Figure 6.** Net fluxes of (a) ethene, (b) propene, (c) butene and (d) isoprene, based on REA (symbols) and gapfilled with ANN (lines). Measurements of (e) air temperature and cumulative precipitation and (f) PAR and net radiation. Eddy covariance measurements of (g) net $CO_2$ flux (h) water vapor flux and (i) sensible heat flux.





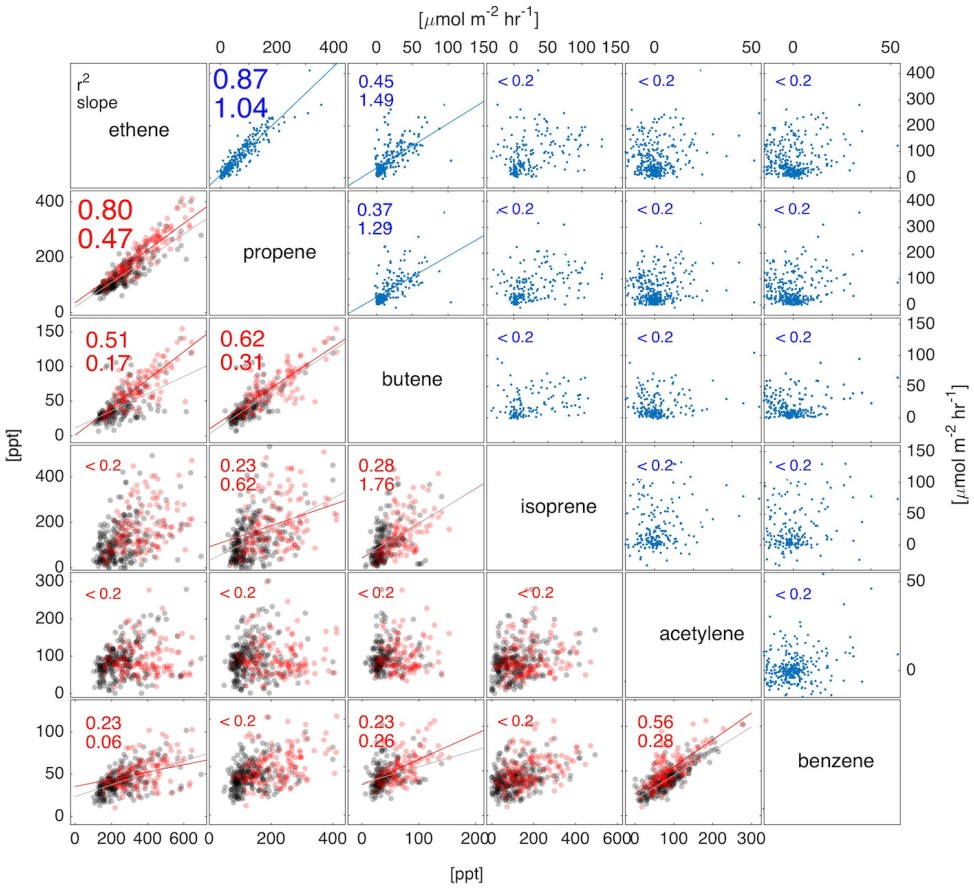

**Figure 7.** Correlation matrix of alkene fluxes (blue), daytime concentrations (red) and nighttime concentrations (black). Linear least squares fit results in colored numbers in top left corner of each plot: $r^2$ (top row) and slopes (bottom row).





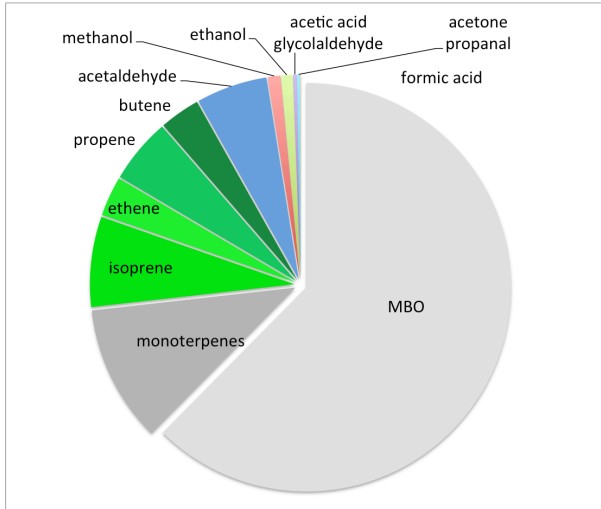

**Figure 8.** Relative OH reactivity for the major known BVOCs emitted at Manitou Experimental Forest, scaled to observed fluxes.

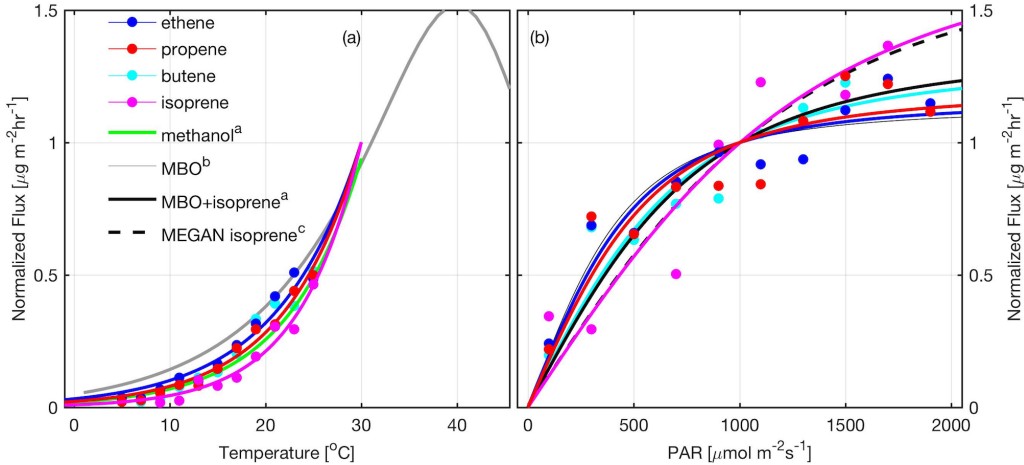

5   **Figure 9.** Parameterized response curves (solid lines) of alkene fluxes bin-averaged in 2 °C classes (circles). Response curves have been normalized to a flux of 1 at (a) reference temperature of 30 °C and (b) reference PAR of 1000 μmol m$^{-2}$ s$^{-1}$. The green, gray and black lines represent the normalized temperature and light response curves from the following references: [a]canopy scale (eddy covariance) PTR-MS (Kaser et al., 2013a), [b] leaf scale Harley *et al.* (1998), [c]MEGAN 2.1 (Guenther et al., 2012).