# Peer review of "Ethene, propene, butene and isoprene emissions from a ponderosa pine forest measured by Relaxed Eddy Accumulation"

_Atmospheric Chemistry and Physics, 2017_

## Referee Comment (RC1) · Anonymous Referee #1 · 12 Jul 2017

In this study, the authors measured and analysed the concentrations and fluxes of light alkenes (ethene, propene, butene), isoprene, acetylene and benzene above and within a ponderosa pine forest canopy during the summer in 2014. They concluded that these light alkenes were originated from local forest canopy, and the measured fluxes of them were about 1 to 3 times as large as the results in previous studies. The authors also found the strong responses of alkene fluxes to temperature and PAR, and then parameterized the responses with two fit functions which agreed well with the measured data.

This study provides a new dataset of measured net fluxes of light alkenes in the am-

bient air and new parameters to model the biogenic emissions of light alkenes. The measurement techniques are described in a clear and detailed way. The parameterization methods will improve or potentially increase the accuracy of biogenic emission models (e.g., MEGAN) in simulating light alkenes. In general, this study has provided a solid understanding of biogenic emissions of light alkenes. Therefore, I recommend to accept this manuscript after some minor revisions.

P6, L1: "in (Businger and Oncley, 1990)" –> "in Businger and Oncley (1990)"

P8, L7: "next Additional" –> "next. Additional"

P9, L26: "correlation coefficients" –> "coefficients of determination (rˆ2)" Here rˆ2 is usually named as the coefficient of determination which represents the percentage of explained variation.

P11, L27: The acronym of quality control QC is not defined before.

P12, L9-10: The names of isoprene, acetylene, benzene need to be added in the table caption.

P12, L17-18: "Storage occurred at night (19-24 MST), offsetting for additional ∼25 % of measured nighttime respiration." This sentence is not clear and needed to be rephrased.

P13, L16-22: The correlations of concentrations and fluxes for acetylene and benzene should be described here or at other places since they are also plotted in Fig. 7.

P14, L11: "BVOC models" –> "BVOC emission models"

P18, L12: The equation 4 is not described clearly. Should the response equation be: F(T) = alpha*exp(beta*(T - 30)) ? Is the actual emission rate calculated as: F = F(T)*F_30? Here F_30 is from Table 2. But if this is true, the actural flux at T=30 [degC] for ethene is: 302.9 * 0.0325 = 9.84 [ug m-2 h-1], which is much lower than the measured flux shown in Table 1.

P18, L9: Is the actual emission rate calculated as: F = F(PAR)*F_1000? Here F_1000 is from Table 2. The same symbol "alpha" is used for both equations 4 and 5, but it represents different parameters. Two different symbols should be used here.

P18, L29-30: "This is consistent with respect to the observed source of isoprene in the understory, which is partially shaded." The meaning of this sentence is not clear. Why the partially-shaded subcanopy should be a large emission source of isoprene?

P18, L31: "... on average a factor of 3.4 smaller than tower-based ..." This is a bit confusing, maybe using percentage is better here.

P18-19: A general comment on section 5.4 and 5.5. 1. In section 5.4, you show the temperature response curves of light alkenes fit the measurement data more than the light response curves. Do you think the equation 4 can be used to simulate the alkene emission rates alone? If so, LDF in MEGAN 2.1 for light alkenes should be set to 0, while currently LDF of ethene is 0.8 and LDF of propene and butene is 0.2. Could you make a solid conclusion in this section about what kind of parameterization methods are better to use in the emission models?

2. In section 5.5, you have compared several parameters between MEGAN 2.1 and the parameteization methods used in this study. Although this manuscript is mainly about measurement and data analysis, I still recommend to implement the new parameters into MEGAN 2.1 and show quantitatively how much more light alkenes are emitted compared to the old version, if this is not difficult to realize.

P19, L9: "CL1 = 0.0007 $\alpha$ = 1.73 in this study, CL1 = 0.0007 $\alpha$ = 1.74 in MEGAN 2.1" This is not consistent with the values shown in Table 2.

P20, L10: "compounds be" –> "compounds should be"

P31: Figure 4: What do the dots mean in the butene plot from day 215 to 220?

P34: Figure 7: The names of acetylene, benzene should be added the caption.

---

## Referee Comment (RC2) · J. Rudolph (Referee) · 1 Aug 2017

The paper presents studies of emissions of light alkenes from a pine forest. Since data on emissions of light alkenes from vegetation are rare and existing emission data have substantial uncertainty the presented information creates new, useful insight into the role of vegetation as source of volatile organic compounds (VOC), a subject that is highly relevant for ACP. The emission studies are based on state of the art methodology, Relaxed Eddy Accumulation (REA). The results are of very high quality, the methodology is clearly explained and overall the discussion is sound and the conclusions justified. The paper is well structured and written, the figures and tables overall of

very good quality. Consequently, the paper merits publication in ACP, although I have a few suggestions for changes and additions to improve the paper.

The two main points are:

i) Gap-filling: I agree that gap-filling can provide a better estimate for averaged monthly or daily fluxes. However, gap-filling, independent of the interpolation procedure, has serious limitations.

- It cannot compensate for bias in the data set resulting from experimental limitations, for example flux data below the DL will be absent from the measured data and gap-filling cannot compensate for this. Indeed, in Figure 6 it seems that for ethene and propene the gap-filled data are always, even at night, above zero. In Figure 9 the PAR dependence of the fluxes predicts that the fluxes are zero for PAR=0.

- When using interpolated data, it is very difficult to derive meaningful statistical criteria. I assume that $\pm$ in Table 1 indicates the standard deviation $\sigma$. I am not sure how to interpret a standard deviation for an interpolated data set. Based on basic statistics the error of the mean can be calculated from $\sigma$ and the number of data points N: Error of mean = $\sigma$/sqrt(N-1). Since there is no limit in the number of interpolated data points this would imply that the error of the mean interpolated flux is effectively zero. Strictly speaking the modeled flux is a calculated value and the error of a calculated value can be determined using (Gaussian) error propagation, which is close to impossible for an ANN interpolated data set.

- Independent of the interpolation procedure, the results of the interpolation can be biased by the assumptions required for the interpolation. Here the situation is especially complex due to the two step interpolation procedure. It is stated that for the ANN gap-filling "inputs were normalized". This needs more explanation. Moreover, from page 9, line 17-18, it seems that the ANN input set was created by interpolation of measured data, although no detail of the interpolation procedure is given.
- An interesting scientific question is: Do the ANN based fluxes provide a better fit to the measured fluxes than a calculation based on the T and PAR dependence developed in 5.5 (Figure 9). This could provide evidence that modeling alkene fluxes using T and PAR alone is insufficient.

ii) Statistics:

- The authors should provide a more detailed evaluation of the significance of the findings. For example, there are no error bars for the binned data in Figure 9. There are no uncertainties for the fitted parameters in Table 2. Based on the scatter of binned data in Figure 9 (especially for the PAR dependence) I would expect substantial uncertainty. This is mentioned in the discussion, but still does not allow an estimate of uncertainty for fluxes calculated from the models. It seems from Figure 6 that the night-time fluxes for ethene and propene are above zero, which is in contrast to the PAR dependence derived in 5.5, the fit parameters given in Table 2 and the fitted functions in Figure 9. Is this a real discrepancy between measured and modeled fluxes (which would indicate shortcomings in the modeled fluxes) or can this be explained by the uncertainty of averages, fits, and binned data, or is this a result of gap-filling?

- On Page 13, lines 27-29 it is mentioned that "The understory REA measurements showed detectable consumption overall for ethene, propene and butane...". However, due to the large uncertainty of the negative fluxes and the small number of data points I am not sure if these fluxes are below zero at a meaningful significance level.

- From Figure 1S it is evident that the probability distribution of measured fluxes cannot be described by a Normal Distribution as implied by the mean values and standard deviations in Table 1. In this case non-parametric statistics (median, percentiles) provide a more realistic insight into the actual distributions than mean and standard deviation or error of mean.

- Correlations: Most of the correlations and linear regressions are for data sets with substantial seemingly random variability of y and x values. Standard linear regression

does not consider uncertainty for the independent variable and an arbitrary identification of the independent variable may result in biased estimates of slope, intercept or R2. A better indicator for the quality of a correlation is the Pearson product moment correlation coefficient, which does not require distinction between a dependent and independent variable. I also think that correlations with R2 values of 0.5 or less do not qualify as good or high correlations. Figure 7 only shows the slopes (without errors) of the linear regressions, not intercepts. Are the axis intercepts zero (within their uncertainty) or was a regression used that did not allow a non-zero axis intercept?

Some details:

Introduction: This is a good, detailed overview of the role and sources of light alkenes in the atmosphere with emphasis on vegetation. I understand that the different estimates etc. represent the uncertainty in current emission inventories and differences between different studies. However, it is not easy for the reader to extract an overall perspective of the role of alkene emissions from vegetation in comparison to other sources of light alkenes or emissions of other VOC from vegetation. A table (or graph) summarising the emission rates of light alkenes from various sources would be very useful for the reader and make it easier for the reader to understand the potential importance of reducing uncertainties in emission rates of light alkenes from vegetation.

Page 5, line 5: "with a lapse between. . ." should be rephrased (gap?).

Page 6, line 19 and other occurrences: "sonic temperature" should be rephrased.

Page 8, line 3 and S, page 5, line 20-30: I accept that, due to the absence of an extreme change between consecutive measurements, alternating the sequence of analysis between the "up bags" and "down bags" allows excluding that storage has a dominating impact on measured fluxes. However, due to the substantial variability of the measured fluxes and the use of two sets of bags, it is not obvious that the flux measurements are entirely free from impact by storage. The authors should present some quantitative estimate at which level of "storage bias" the mentioned "seesaw" pattern (pages 5, line

27-28) would clearly be visible.

Page 8, line 15-25, Fig. S1: Quality control, detection limits and flagged data. "Quality control for each hourly REA flux measurement was checked against eight potential flags associated with the sample volumes, meteorological conditions or footprint analysis". There is an explanation in the supplement. However, since 47% of used data were flagged for not strictly meeting all criteria some quantitative estimate should be given to which extent this may impact data quality. Data flagged with more than 4 flags or data not meeting specific footprint criteria are excluded. I am not sure about the rational for this specific threshold. Moreover, it seems that a large portion of very low fluxes were flagged or failed QC. How does this impact the overall averages and representativeness of the flux data? How were flux data below the lowest flux detection limit treated in further data evaluation? The way data below the LDL are used is important since this may create bias for averages, fits, modeling etc. It is also mentioned that some fluxes were negative. How were they used in the data set? From Figure 7 it seems that there are no negative fluxes, but it is mentioned that a small number of fluxes which met QC are negative.

Page 8, 9, S2, Understory REA fluxes: I understand that measuring the understory flux is very difficult, especially since it seems that the flux is small. Nevertheless, the very small fraction of measurements that passed QC raises the question of representativeness of the few good quality data. Since the main result of this study is that understory fluxes are (on average) small compared to the total flux, rejected data which could be used to provide meaningful upper limits of understory fluxes may strengthen this point.

5.2 and 5.3: Both subchapters present a comparison between literature and this study. Combining both chapters would allow a more consistent comparison. For example, in 5.2 the alkene fluxes are placed in context to $CO_2$ and other VOC fluxes. However, in 5.3 this context is not considered. Yet, it would be very important to understand if some of the differences in alkene fluxes maybe related to differences in assimilation or respiration rates and so on.

[Figure]

Page 16, lines 25-27: "We utilized fluxes instead of concentrations to provide a measure of OH reactivity that is independent of elevated concentrations associated with pollution events and more representative of site specific sources." This creates a distorted view of the importance of emissions. The atmospheric OH-reactivity of a VOC at a given concentration ([VOC]) is determined by [VOC] kOH, as mentioned in the previous sentence and the cited literature. In a simplified steady state [VOC] is proportional to the flux and inversely proportional the reactivity (kOH). Consequently, the overall relative impact of an emission is simply determined by the emission flux. The reactivity determines the temporal and spatial scales at which this happens. It should also be considered that alkenes also react with ozone, OH reactivity therefore only determines one part of the overall alkene reactivity.

Page 17, lines 11-12: Based on the uncertainties of averaged fluxes, extrapolation etc. I am not certain that a 20-60% difference is really significant. I also do not understand why the estimate of the seasonal flux average is based on such a simple extrapolation. The paper presents two more detailed models (gap-filling based on ANN and the T and PAR dependence presented in 5.4) which can be used to calculate averaged fluxes for comparison.

Combining subchapters 5.4 and 5.5 would allow streamlining the discussion of flux parametrizations and model predictions, for example how well MEGAN 2.1 parameterizations agree with the measured alkene (and not only isoprene) fluxes. Based on the current discussion it is not evident that (and why) "Modifying the light and temperature parameterizations for light alkenes in the vegetation emissions model will lead to a corresponding increase in estimated global emissions for these compounds".

Conclusions: They are more or less a summary. I agree with the overall conclusions that the current understanding of alkene emissions from vegetation is insufficient and that alkene emissions from vegetation can be relevant for the chemistry of the atmosphere. However, based on the many interesting aspects presented and discussed in the paper I am a bit disappointed that the suggestions for tackling open questions

and reducing uncertainties is basically a generic "more research needed" approach. Specifically, the very good correlation between ethene and propene fluxes is striking and raises questions about origin and the factors determining the emissions of these two alkenes.

---

## Author Comment (AC1) · 4 Sep 2017

acp-2017-363: Ethene, propene, butene and isoprene emissions from a ponderosa pine forest measured by Relaxed Eddy Accumulation

**RESPONSE TO REVIEWER #1 (Anonymous Referee #1)**

**We thank reviewer 1 for the thorough and positive review of our manuscript and especially for the helpful comments, which we have incorporated in the revised manuscript. We believe that this revised manuscript is improved in quality and clarity. In the following text, we address each comment individually and highlight changes to the revised manuscript. Reviewer 1 comments appear in italics, and our responses are in bold type-face.**

*In this study, the authors measured and analysed the concentrations and fluxes of light alkenes (ethene, propene, butene), isoprene, acetylene and benzene above and within a ponderosa pine forest canopy during the summer in 2014. They concluded that these light alkenes were originated from local forest canopy, and the measured fluxes of them were about 1 to 3 times as large as the results in previous studies. The authors also found the strong responses of alkene fluxes to temperature and PAR, and then parameterized the responses with two fit functions which agreed well with the measured data.*

*This study provides a new dataset of measured net fluxes of light alkenes in the ambient air and new parameters to model the biogenic emissions of light alkenes. The measurement techniques are described in a clear and detailed way. The parameterization methods will improve or potentially increase the accuracy of biogenic emission models (e.g., MEGAN) in simulating light alkenes. In general, this study has provided a solid understanding of biogenic emissions of light alkenes. Therefore, I recommend to accept this manuscript after some minor revisions.*

*P6, L1: "in (Businger and Oncley, 1990)" –> "in Businger and Oncley (1990)"*

**Corrected.**

*P8, L7: "next Additional" –> "next. Additional"*

**Corrected.**

*P9, L26: "correlation coefficients" –> "coefficients of determination ($r^2$)" Here $r^2$ is usually named as the coefficient of determination which represents the percentage of explained variation.*

**We agree and adopted the proposed terminology.**

*P11, L27: The acronym of quality control QC is not defined before.*

**Rephrased to: "the reported quality-ensured fluxes"**

*P12, L9-10: The names of isoprene, acetylene, benzene need to be added in the table caption.*

**Corrected.**

*P12, L17-18: "Storage occurred at night (19-24 MST), offsetting for additional ~ 25% of measured nighttime respiration." This sentence is not clear and needed to be rephrased.*

**We thank Reviewer 1 for pointing this out.  This has been rephrased to: "Storage occurred at night (19-24 MST), leading to an underrepresentation in measured night time respiration on the order of ~25 %."**

*P13, L16-22: The correlations of concentrations and fluxes for acetylene and benzene should be described here or at other places since they are also plotted in Fig. 7.*

**We have added a correlation analysis for acetylene and benzene in section 4.4 and have added text in section 5.1 as follows: "In contrast, no such diurnal patterns in concentration are observed for the primarily anthropogenic compounds (acetylene and benzene), and their fluxes are near zero (Table 1). Consequently, correlations between the light alkenes and either acetylene or benzene are poor (concentrations) or non-significant (fluxes)."**

*P14, L11: "BVOC models" –> "BVOC emission models"*

**Corrected.**

*P18, L12: The equation 4 is not described clearly. Should the response equation be: F(T) = alpha\*exp(beta\*(T - 30)) ? Is the actual emission rate calculated as: F = F(T)\*F_30? Here F_30 is from Table 2. But if this is true, the actual flux at T=30 [degC] for ethene is: 302.9 \* 0.0325 = 9.84 [ug m-2 h-1], which is much lower than the measured flux shown in Table 1.*

**Equation 4 was originally intended to be a normalization factor to the observed flux at 30$^{o}$C.  This would allow the curve fits for different gases to be compared more easily by being plotted on a single graph (i.e., temperature = 30$^{o}$C and F(T) = 1 for all gases).  For example:**

**>> 0.0325 x exp(0.1142 x 30)  =  1**

**To scale to meaningful flux units (F), however, it was indeed necessary to multiply by the reference flux (F$_{30}$):    F = F(T) \* F$_{30}$.  We see how this was unclear, so in the revised manuscript (eq.4 is now eq.5), the equation was changed slightly, to improve clarity and to conform better with the MEGAN nomenclature.  The equation is now written as:**

$$F(T_{LDIF}) = F_{ref} * \exp\big(\beta(T - T_{ref})\big) \qquad \text{(Eq 5)}$$

**Here, F(T$_{LDIF}$) describes the light independent fraction of temperature response, where $\beta$ is an empirical coefficient (Table 2), T is the ambient temperature (°C), T$_{ref}$ is constant = 30 °C, and F$_{ref}$ is the observed flux at 30 °C.**

**The normalization by F$_{ref}$ is now included as part of the equation, and fitted coefficients from Table 3 can be plugged in to calculate fluxes in units [μg m$^{-2}$ hr$^{-1}$]. For example, at 30°C with reference flux for ethene (table 3) being 316 [μg m$^{-2}$ hr$^{-1}$]:**

**F(30°C)  = 316 \* exp(0.114\*(30-30))  = 316, as expected.**

acp-2017-363: Ethene, propene, butene and isoprene emissions from a ponderosa pine forest measured by Relaxed Eddy Accumulation

*P18, L9: Is the actual emission rate calculated as: F = F(PAR)\*F_1000? Here F_1000 is from Table 2. The same symbol "alpha" is used for both equations 4 and 5, but it represents different parameters. Two different symbols should be used here.*

**The parameterization was indeed a normalization function to $F_{1000}$. We reassigned the Greek symbols in the equations so that each letter is now used only once. Similar to the temperature response function, we have clarified the PAR response equation (eq. 4) to improve clarity. Now the normalization by $F_{1000}$ is included in the function, as follows:**

$$F(PAR) = \frac{\alpha\, C_{L1}\, PAR}{\sqrt{1 + \alpha^2 PAR^2}} * F_{1000} \tag{4}$$

*P18, L29-30: "This is consistent with respect to the observed source of isoprene in the understory, which is partially shaded." The meaning of this sentence is not clear. Why the partially-shaded subcanopy should be a large emission source of isoprene?*

**We acknowledge that this sentence was unclear. The canopy consists almost entirely of mature ponderosa pine trees, which are considered to be low isoprene emitters. The plant composition in the understory includes juvenile coniferous trees, grasses and ferns, and a considerable amount of the ground surface is either bare ground or leaf and needle litter. During our understory case study, measured fluxes above the forest ground were a similar magnitude as, or larger than, fluxes measured above the forest canopy. Thus, we hypothesized that almost the entire isoprene source was below the forest canopy.**

**However, the PAR values that we used in the light-response curve computations were measured above the forest canopy. For compounds that are presumably produced by the canopy (e.g., ethene/propene/butene), the fluxes flatten out at PAR > 1000, whereas isoprene fluxes still increase with increasing light. This was what we sought to explain.**

**Since only a fraction of the light observed above the canopy penetrates to the forest ground, the forest ground experiences a muted sunlight intensity, and therefore the subcanopy may not experience the high sunlight intensity where fluxes start to asymptote.**

**We rephrased the sentence as follows (section 5.4):**
**"It should be noted that the PAR measurements employed to compute the light-response curves were measured above the canopy, while the observed source of isoprene appears to be in the vegetated understory, which experiences more diffuse light. In fact, PAR intensity measured near ground level (2 m a.g.l.) was on average 50 ± 30 % (standard deviation) of the measured PAR above the forest canopy. Hence, the subcanopy isoprene source(s) may experience an optimum quantum yield at much larger incident PAR (measured above canopy) than the light alkene source within the ponderosa pine canopy, explaining the different light response curve."**

*P18, L31: "... on average a factor of 3.4 smaller than tower-based ..." This is a bit confusing, maybe using percentage is better here.*

acp-2017-363:  Ethene, propene, butene and isoprene emissions from a ponderosa pine forest measured by Relaxed Eddy Accumulation

**We changed this to a percentage [%], see above.**

*P18-19: A general comment on section 5.4 and 5.5.*
*1. In section 5.4, you show the temperature response curves of light alkenes fit the measurement data more than the light response curves. Do you think the equation 4 can be used to simulate the alkene emission rates alone?*

**We explored this by predicting alkene fluxes in 5 different ways: a) only eq. 4 (PAR); b) only eq. 5 (temperature, light independent); c) only eq. 6 (T, light dependent); d) the weighted (according to the LDF in MEGAN) combination of eq 5 / 6; and e) our initial gap-filling method (Artificial Neural Networks). Here are the statistics as root mean square error (RMSE) and Pearson's correlation coefficient ($\rho$) of comparison between predicted and measured fluxe:**

**Q&A Table 1.**

|  | ethene | | propene | | butene | | isoprene | |
|---|---|---|---|---|---|---|---|---|
|  | RMSE | $\rho$ | RMSE | $\rho$ | RMSE | $\rho$ | RMSE | $\rho$ |
| F(PAR) | 46.5 | 0.68 | 42.2 | 0.69 | 16.3 | 0.7 | 41.4 | 0.54 |
| F(T_LIDF) | 48.4 | 0.72 | 46 | 0.68 | 16.8 | 0.64 | 44.3 | 0.48 |
| F(T_LDF) | 45.7 | 0.72 | 48 | 0.69 | 16.9 | 0.65 | 41.2 | 0.56 |
| F(T_MEGAN) | 45.5 | 0.73 | 43.1 | 0.69 | 16.5 | 0.68 | 41.2 | 0.56 |
| ANN | 32.1 | 0.83 | 27.7 | 0.8 | 8.6 | 0.8 | 28.6 | 0.64 |

**Yes, fluxes can be reasonably predicted solely as a function of T or PAR.  However, the combined approaches F(T_MEGAN) and ANN yield better predictions of fluxes. Since all studied light alkenes show a response to light, we believe light should be included in parameterizations. If one wanted to predict fluxes solely on one equation, one should use the temperature response function with best fitting results, and this is now presented in table 3 of the revised manuscript.**

*If so, LDF in MEGAN 2.1 for light alkenes should be set to 0, while currently LDF of ethene is 0.8 and LDF of propene and butene is 0.2. Could you make a solid conclusion in this section about what kind of parameterization methods are better to use in the emission models?*

**Our results demonstrate that ethene and propene fluxes are highly correlated, and we propose to change the LDF of propene to 0.8, matching ethene.**

*2. In section 5.5, you have compared several parameters between MEGAN 2.1 and the parameterization methods used in this study. Although this manuscript is mainly about measurement and data analysis, I still recommend to implement the new parameters into MEGAN 2.1 and show quantitatively how much more light alkenes are emitted compared to the old version, if this is not difficult to realize.*

**This is a very worthwhile exercise, but we feel this is more appropriate in the context of a modeling paper that can explore the implications of these new equations better.  We prefer to keep the focus of this manuscript on the measurement techniques, novel**

acp-2017-363: Ethene, propene, butene and isoprene emissions from a ponderosa pine forest measured by Relaxed Eddy Accumulation

**results, and improved flux parameterizations, all of which should set up a future modeling paper nicely.**

*P19, L9: "CL1 = 0.0007*
*α = 1.73 in this study, CL1 = 0.0007*
*α = 1.74 in MEGAN 2.1"*
*This is not consistent with the values shown in Table 2.*

**Thanks for spotting that mistake. The values in the text now correspond with values in table 3. The table version was correct and remained unchanged.**

*0: "compounds be" –> "compounds should be"*

**Corrected.**

*P31: Figure 4: What do the dots mean in the butene plot from day 215 to 220?*

**We had trouble locating any wayward dots. The acetylene and benzene subplot had a blue and red dot used in the legend, which were perhaps too subtle. We changed the legend and hope to have resolved the issue.**

*P34: Figure 7: The names of acetylene, benzene should be added the caption*

**Corrected.**

---

## Author Comment (AC2) · 4 Sep 2017

acp-2017-363: Ethene, propene, butene and isoprene emissions from a ponderosa pine forest measured by Relaxed Eddy Accumulation

**RESPONSE TO REVIEWER #2 (Jochen Rudolph)**

**Dr. Rudolph provided multiple suggestions on how to clarify and further improve the manuscript, and we address each of them below. We appreciate Dr. Rudolph for his detailed and thorough (and positive) review, and the revised manuscript incorporates improvements in readability, clarity, statistical analysis and emphasis of the study impact. Reviewer comments appear in italics, and our responses are in bold**

*The paper presents studies of emissions of light alkenes from a pine forest. Since data on emissions of light alkenes from vegetation are rare and existing emission data have substantial uncertainty the presented information creates new, useful insight into the role of vegetation as source of volatile organic compounds (VOC), a subject that is highly relevant for ACP. The emission studies are based on state of the art methodology, Relaxed Eddy Accumulation (REA). The results are of very high quality, the methodology is clearly explained and overall the discussion is sound and the conclusions justified. The paper is well structured and written, the figures and tables overall of very good quality. Consequently, the paper merits publication in ACP, although I have a few suggestions for changes and additions to improve the paper. The two main points are:*

*i) Gap-filling: I agree that gap-filling can provide a better estimate for averaged monthly or daily fluxes. However, gap-filling, independent of the interpolation procedure, has serious limitations. - It cannot compensate for bias in the data set resulting from experimental limitations, for example flux data below the DL will be absent from the measured data and gap-filling cannot compensate for this. Indeed, in Figure 6 it seems that for ethene and propene the gap-filled data are always, even at night, above zero. In Figure 9 the PAR dependence of the fluxes predicts that the fluxes are zero for PAR=0.*

**This is correct: the ANN predictions are > 0 for PAR = 0. We will discuss later how the PAR dependent equation is not the optimal equation to use to determine fluxes, owing in part to this noted shortcoming.**

*- When using interpolated data, it is very difficult to derive meaningful statistical criteria. I assume that ± in Table 1 indicates the standard deviation $\sigma$ . I am not sure how to interpret a standard deviation for an interpolated data set. Based on basic statistics the error of the mean can be calculated from $\sigma$ and the number of data points N: Error of mean = $\sigma$ /sqrt(N-1). Since there is no limit in the number of interpolated data points this would imply that the error of the mean interpolated flux is effectively zero. Strictly speaking the modeled flux is a calculated value and the error of a calculated value can be determined using (Gaussian) error propagation, which is close to impossible for an ANN interpolated data set.*

**That is also correct: Table 1 showed the average flux ± sd. The ANN-derived fluxes (i.e., gap-filled results) were compared against the validated observed fluxes to assess if there was any bias associated with the observational gaps. To do this, hourly fluxes were extracted from the ANN model from days 174 to 225 (n=1223). The mean and standard deviation were derived from these values. In light of comments below, we have replaced Table 1 means and standard deviations with the median and 10[th]/90[th] percentile ranges. In case readers still wish to see the average ± sd, we moved these values to the Supplementary Material, Table S2.**

*- Independent of the interpolation procedure, the results of the interpolation can be biased by the assumptions required for the interpolation. Here the situation is especially complex due to the two step interpolation procedure. It is stated that for the ANN gap-filling "inputs were normalized". This needs more explanation.*

**The reviewer is referring to Section 3.5 "Prior to gap filling, inputs were normalized and gap-filled with average values from the surrounding days..."  Specifically, inputs were scaled to values from -1 (for the minimum observation) to +1 (for maximum observation). This procedure is commonly used in machine learning, when input variables have different scales/units. The equation used was: $y = (y_{max}-y_{min})*(x-x_{min})/(x_{max}-x_{min}) + y_{min}$.  The normalization procedure is clarified in the revised manuscript in Section 3.5 as follows: "Prior to gap filling, input variables were normalized on a scale of -1 (for minimum value) to +1 (for maximum value)."**

*Moreover, from page 9, line 17-18, it seems that the ANN input set was created by interpolation of measured data, although no detail of the interpolation procedure is given.*

**For the ANN algorithms, explanatory variables are required to be continuous (i.e., without gaps). The few time periods when input drivers were missing were programmed to be filled with average values from the surrounding days (i.e, the same time on the next day) or through a simple 2-D interpolation. It turns out that there were no gaps in PAR and only 1% missing values (n=16 of 1223) for $H_2O$ flux, temperature and std deviation of wind speed. Thus, we removed this line to avoid unnecessary confusion.  (The entire input variable dataset had 2472 observations each, but included time periods outside the REA flux measurement period, so that also was adjusted accordingly).**

*An interesting scientific question is: Do the ANN based fluxes provide a better fit to the measured fluxes than a calculation based on the T and PAR dependence developed in 5.5 (Figure 9). This could provide evidence that modeling alkene fluxes using T and PAR alone is insufficient.*

**See response to reviewer 1.  To reiterate:  we explored this by predicting alkene fluxes in 5 different ways: a) only eq. 4 (PAR); b) only eq. 5 (temperature, light independent); c) only eq. 6 (T, light dependent); d) the weighted (according to the LDF in MEGAN) combination of eq 5 / 6; and e) our initial gap-filling method (Artificial Neural Networks). Here the statistics are reported as root mean square error (RMSE) and Pearson's correlation coefficient ($\rho$) of comparison between predicted and measured fluxes**

|            | ethene |      | propene |      | butene |      | isoprene |      |
|------------|--------|------|---------|------|--------|------|----------|------|
|            | RMSE   | $\rho$ | RMSE  | $\rho$ | RMSE | $\rho$ | RMSE    | $\rho$ |
| F(PAR)     | 46.5   | 0.68 | 42.2    | 0.69 | 16.3   | 0.7  | 41.4     | 0.54 |
| F(T_LIDF)  | 48.4   | 0.72 | 46      | 0.68 | 16.8   | 0.64 | 44.3     | 0.48 |
| F(T_LDF)   | 45.7   | 0.72 | 48      | 0.69 | 16.9   | 0.65 | 41.2     | 0.56 |
| F(T_MEGAN) | 45.5   | 0.73 | 43.1    | 0.69 | 16.5   | 0.68 | 41.2     | 0.56 |
| ANN        | 32.1   | 0.83 | 27.7    | 0.8  | 8.6    | 0.8  | 28.6     | 0.64 |

**According to these results, the ANN-based fluxes provide a superior fit to the measured fluxes overall than the calculations based on Temperature (LIDF or LDF) and PAR. However, the combined T and PAR results perform almost as well. We see high potential in ANN as a gap-filling tool. We now include the text: "ANN is increasingly used in eddy covariance studies because of its ability to resolve non-linear relationships and complex interactions between flux drivers (Dengel et al., 2013, Papale and Valentini, 2003)"**

**That being said, we do believe that a more mechanistic oriented model (such as MEGAN) is the better choice to predict data for sites or times where no measurements are available (which is necessary for training in the machine learning approach). Both approaches have their relevance and we are applying both where appropriate.**

**Additional references added to manuscript:**
**Dengel et al 2013: Testing the applicability of neural networks as a gap-filling method using CH4 flux data from high latitude wetlandsBiogeosciences. doi:10.5194/bg-10 8185-2013**
**Papale and Valentini 2003: A new assessment of European forests carbon exchanges by eddy fluxes and artificial neural network spatialization. Global Change Biology. (9) 525-535**

*ii) Statistics:*
*- The authors should provide a more detailed evaluation of the significance of the findings. For example, there are no error bars for the binned data in Figure 9. There are no uncertainties for the fitted parameters in Table 2. Based on the scatter of binned data in Figure 9 (especially for the PAR dependence) I would expect substantial uncertainty. This is mentioned in the discussion, but still does not allow an estimate of uncertainty for fluxes calculated from the models.*

**We have calculated 10-90$^{th}$ percentiles to the binned fluxes in figure 9, now illustrated on the plots. We also provide 90$^{%}$ confidence bounds for the fitting coefficients of Equation 4 in Table 2 and of Equation 5 in Tables 3 and S4.**

*It seems from Figure 6 that the night-time fluxes for ethene and propene are above zero, which is in contrast to the PAR dependence derived in 5.5, the fit parameters given in Table 2 and the fitted functions in Figure 9. Is this a real discrepancy between measured and modeled fluxes (which would indicate shortcomings in the modeled fluxes) or can this be explained by the uncertainty of averages, fits, and binned data, or is this a result of gap-filling?*

**Our measurements do indicate small emissions of propene and ethene for PAR = 0 ($\approx$ 20 µg m$^{-2}$ hr$^{-1}$), with the caveat that only a few of these emission rates exceed flux detection limits (n < 10). This contrasts the PAR dependent equation, in which the calculated flux goes to zero as PAR goes to zero (i.e., PAR is in the numerator). The light and temperature response curves have only been fitted to measured REA flux data, without gap filling the dataset. Thus this discrepancy appears to be a shortcoming in the PAR-dependent equation, not a consequence of gap-filling or binning results.**

**There is other evidence of nighttime positive fluxes for the light alkenes and other BVOCs. The diurnally averaged fluxes of light alkenes at Harvard Forest (Goldstein *et al*.**

**1996), the only directly comparable net ecosystem flux measurements, suggest that most fluxes were positive, even around midnight, which is in line with our results. Non-zero, positive fluxes around midnight are also apparent in fluxes for methanol and ethanol (Schade and Goldstein, 2001) and monoterpenes (Kaser *et al.*, 2013). [All references in the original manuscript].**

**These results suggest that it may be necessary to add a baseline term (ε) to eq. 5 (PAR response curve) for several BVOCs, to account for non-zero fluxes at PAR = 0, such as:**

$$F(PAR) = \varepsilon + \left[\frac{\gamma\, C_{L1}\, PAR}{\sqrt{1 + \gamma^2 PAR^2}}\right]$$

**However, adjusting the PAR model equation is outside the scope of the current manuscript, and we do not feel this is particularly useful for this study, as the temperature dependent equations appear to address the shortcomings of the PAR dependent equation.**

*- On Page 13, lines 27-29 it is mentioned that "The understory REA measurements showed detectable consumption overall for ethene, propene and butane...". However, due to the large uncertainty of the negative fluxes and the small number of data points I am not sure if these fluxes are below zero at a meaningful significance level.*

**The phrase "detectable consumption" means that the uptake rates for non-flagged REA measurements were larger than detection limits. We acknowledge that conclusions based on the understory fluxes have larger uncertainties than the above canopy REA measurements owing to the relatively sparse data and possible violations of Monin-Obukhov similarity theory (which we tried to minimize to our best abilities, see Section 3.4 and Fig S2). That being said, it does appear that the uptake rates for ethene, propene, and maybe butene are indeed significant, at least for this one day. We have now added Figure S3 to the supplementary information.**

*- From Figure 1S it is evident that the probability distribution of measured fluxes cannot be described by a Normal Distribution as implied by the mean values and standard deviations in Table 1. In this case non-parametric statistics (median, percentiles) provide a more realistic insight into the actual distributions than mean and standard deviation or error of mean.*

**We have changed the flux statistics in Table 1 to report the median and 10th-90th percentiles. Also in the results section 4, the text has been changed to report medians and the inter-quartile (25th and 75th percentile) ranges, rather than mean ± std.**

*- Correlations: Most of the correlations and linear regressions are for data sets with substantial seemingly random variability of y and x values. Standard linear regression does not consider uncertainty for the independent variable and an arbitrary identification of the independent variable may result in biased estimates of slope, intercept or $R^2$. A better indicator for the quality of a correlation is the Pearson product moment correlation coefficient, which does not require distinction between a dependent and independent variable. I also think that correlations with R2 values of 0.5 or less do not qualify as good or high correlations.*

**We agree with these points. We re-calculated the correlations using the Pearson's correlation coefficient and updated figure 7 to indicate this.**

*Figure 7 only shows the slopes (without errors) of the linear regressions, not intercepts.*

**We have also updated figure 7 to show the resulting slopes and intercepts for plots where the Pearson's correlation coefficient >0.5. In addition, the 90% confidence bounds for slopes and intercepts can now be found as a table in the updated supplementary material (Table S2).**

*Are the axis intercepts zero (within their uncertainty) or was a regression used that did not allow a non-zero axis intercept?*

**Intercepts are not forced to be zero. Also see comment above.**

*Some details:*
*Introduction: This is a good, detailed overview of the role and sources of light alkenes in the atmosphere with emphasis on vegetation. I understand that the different estimates etc. represent the uncertainty in current emission inventories and differences between different studies. However, it is not easy for the reader to extract an overall perspective of the role of alkene emissions from vegetation in comparison to other sources of light alkenes or emissions of other VOC from vegetation. A table (or graph) summarising the emission rates of light alkenes from various sources would be very useful for the reader and make it easier for the reader to understand the potential importance of reducing uncertainties in emission rates of light alkenes from vegetation.*

**Such a table exists in Poisson et al., 2000, and we think this may be suitable for a future manuscript with various model results.**

*Page 5, line 5: "with a lapse between ... " should be rephrased (gap?).*

**Changed to "gap"**

*Page 6, line 19 and other occurrences: "sonic temperature" should be rephrased.*

**We chose to retain the term 'sonic temperature', which we consider an accepted terminology for the temperature measured by the sonic anemometer.**

*Page 8, line 3 and S, page 5, line 20-30: I accept that, due to the absence of an extreme change between consecutive measurements, alternating the sequence of analysis between the "up bags" and "down bags" allows excluding that storage has a dominating impact on measured fluxes. However, due to the substantial variability of the measured fluxes and the use of two sets of bags, it is not obvious that the flux measurements are entirely free from impact by storage. The authors should present some quantitative estimate at which level of "storage bias" the mentioned "seesaw" pattern (Suppl. pages 5, line 27-28) would clearly be visible.*

**We thank the referee for encouraging further scrutiny of this statement. It turns out that the seesaw pattern we described in the supplementary text was neither systematic (i.e.,**

**the pattern only occurred occasionally) nor consistent (i.e., the fluctuations when they did appear were not always at 1 hour intervals). Thus, this precautionary statement was a bit overstated. We conclude that there is not a systematic storage issue but that fluctuations in the sunrise and sunset transitions were natural fluctuations in the flux across the canopy. We have adjusted the text in the Supplementary Information as follows: "Seesaw patterns were observed occasionally during the sunrise and sunset transitions, but they were neither systematic (i.e., did not occur regularly) nor consistent (i.e., closer examination shows that fluctuations were not necessarily hourly). In addition, these are periods when ozone concentrations were expected to be low reducing their importance in terms of storage issues. Even under these conditions, negative fluxes were generally not observed."**

*Page 8, line 15-25, Fig. S1: Quality control, detection limits and flagged data. "Quality control for each hourly REA flux measurement was checked against eight potential flags associated with the sample volumes, meteorological conditions or footprint analysis". There is an explanation in the supplement. However, since 47% of used data were flagged for not strictly meeting all criteria some quantitative estimate should be given to which extent this may impact data quality. Data flagged with more than 4 flags or data not meeting specific footprint criteria are excluded. I am not sure about the rational for this specific threshold.*

**We acknowledge that the flagging methodology could use further clarification. We have adjusted the text to reflect that we used 3 tests. The first test was a turbulence assessment, following best practice guides by Foken and Wichura (1996), and Mauder and Foken (2004). Any data, from 30-min intervals with poor turbulence characteristics was critically marked as "bad quality" and discarded from measurements.**
**The second test involved applying a total of 5 REA apparatus specific flags. These were listed in the supplementary material as "REA flags" (QC 1-5, now REA flags a-e). Data getting flagged for 1 or 2 single REA QC check remained in the dataset, flagged as 'medium quality'. If 3 or more REA QC tests were flagged, this was deemed a critical failure, but this only happened sporadically.**
**A third test on homogenous flux footprints was employed (see section 3.7 in the manuscript). If analyses indicated an inhomogeneous flux footprint (e.g., due to the crossing of a road east of the tower), data were non-critically flagged. The employed footprint analysis (Hsieh *et al*., 2000) should only be treated as a proxy/indication of the actual size and spatial distribution of sources and sinks. Hence, we do not believe that this method should be used as a critical flag. In fact, the distance from the tower to the maximum contributing source area (eq. 19, Hsieh *et al*., 2000) is much smaller (by a factor ≥ 10) than the boundaries (90% flux footprint) plotted in figure 3. Regardless of wind direction, the maximum contributing source area lies well within the ponderosa pine forest.**

*However, since 47% of used data were flagged for not strictly meeting all criteria some quantitative estimate should be given to which extent this may impact data quality.*

**We have now clarified that 13% of the data were critically flagged and removed, 18% were flagged as medium quality but not removed, and 16% were flagged for footprint issues but are not likely to be problematic. Quantifying the impact that specific flags have on overall data quality is not straightforward. However, it is clear from Figure S1 that a**

**majority of the flags occur when the measured flux is near zero. Flagged (but accepted) data do not manifest themselves as outliers from the surrounding results; hence, most uncertainties introduced by these flagged data relate to small fluxes, in the absolute sense. Nevertheless, flagging data is an important data quality assessment tool for comparative purposes used in international communities (e.g., FluxNet). Generally, for seasonal and annual timescales, flagged data is treated as acceptable data.**

*Moreover, it seems that a large portion of very low fluxes were flagged or failed QC. How does this impact the overall averages and representativeness of the flux data?*

**We added the following sentence to section 4.2. to address this comment: "Gap filling REA fluxes (Fig. 6) using artificial neural networks (i.e., modeled results) generally increased median emissions (Table 1); however, differences between groups of modeled and observed fluxes were non-significant (anova, α = 0.05) suggesting that the selectivity of quality controlled measurements might lead to only a minor under-prediction of diurnal averages."**

*How were flux data below the lowest flux detection limit treated in further data evaluation? The way data below the LDL are used is important since this may create bias for averages, fits, modeling etc.*

**Flux observations < LDL were discarded for statistical analysis (median, IQR, percentiles) and for any fitting/response analysis. This is now mentioned in the revised manuscript in section 3.3: "Flux observations below $F_{min}$ were excluded from overall statistical analyses (median and percentiles) and for the curve fitting in response to temperature and PAR."**

*It is also mentioned that some fluxes were negative. How were they used in the data set? From Figure 7 it seems that there are no negative fluxes, but it is mentioned that a small number of fluxes which met QC are negative.*

**The frequency of occurrence and magnitude of deposition events are described in the text (Section 4.2). For ethene, 1.3% of quality ensured observations indicated deposition, for propene 2.3%, butene 1.3%, and only for isoprene a larger amount of data has a negative sign at 13%. In Figure 7, we excluded the negative outliers for ethene, propene and butene, and this is now indicated in the caption: "Negative fluxes for the light alkenes (1.3% to 2.3% of the light alkene fluxes) are excluded from the plot and the regression statistics." Also, the temperature response equation (Eq. 4, now Eq 5 in the revised manuscript) does not allow negative predictions. Hence a comparison between modeled and observed fluxes was only made with observed emissions (i.e., positive fluxes). If the few observed deposition events are included in the dataset, median fluxes decrease slightly, for ethene by 1.1, propene by 2.9, and butene by 1.3 ($\mu g\ m^{-2}\ h^{-1}$). No significant difference was found between datasets including negative fluxes as compared to the dataset without negative fluxes. We now include the text: "Negative fluxes were too infrequent and small to be captured in ANN model predictions, and are excluded in the comparison of results (Table 1). Their inclusion would decrease median observed fluxes by 1.1, 2.9, and 1.3 $\mu g\ m^{-2}\ h^{-1}$ for ethene, propene and butene, respectively."**

*Page 8, 9, S2, Understory REA fluxes: I understand that measuring the understory flux is very difficult, especially since it seems that the flux is small. Nevertheless, the very small fraction of measurements that passed QC raises the question of representativeness of the few good quality data. Since the main result of this study is that understory fluxes are (on average) small compared to the total flux, rejected data which could be used to provide meaningful upper limits of understory fluxes may strengthen this point.*

**A total of 6 out of 10 alkene fluxes for understory REA measurements were conducted during acceptable turbulence conditions. We considered the point about including the 4 rejected time points. Including these data would support the main message of how the understory could not account for the light alkene emissions observed above canopy and is also consistent with the isoprene emissions.**

**For ethene, the rejected understory flux results were relatively small (-9.7, -0.8, -5.6 and +3.8 $\mu$g m$^{-2}$ h$^{-1}$) and comparable to the quality-ensured understory measurements reported in Table 1. For propene, two of the rejected fluxes were similar to reported values, but two were large and negative: -98.4, -2.4, -2.6, and -54.4 $\mu$g m$^{-2}$ h$^{-1}$. For butene, all four rejected fluxes are comparable to the reported fluxes: -10.4, -1.8, -7.0, -8.1 $\mu$g m$^{-2}$ h$^{-1}$. For isoprene, the rejected fluxes are small in the morning (5.7, -2.2 and 5.1 $\mu$g m$^{-2}$ h$^{-1}$) while the one rejected flux in the p.m. is very large (266 $\mu$g m$^{-2}$ h$^{-1}$). We have now included the complete set of results in Figure S3 to illustrate that the rejected data are consistent with the overall understory results, and that including them would not yield meaningful improvements to the understory flux statistics.**

*5.2 and 5.3: Both subchapters present a comparison between literature and this study. Combining both chapters would allow a more consistent comparison. For example, in 5.2 the alkene fluxes are placed in context to CO2 and other VOC fluxes. However, in 5.3 this context is not considered. Yet, it would be very important to understand if some of the differences in alkene fluxes maybe related to differences in assimilation or respiration rates and so on.*

**We appreciate the motivation to understand what ecophysiological controls are driving emissions. The purpose of having two sections was in order to clearly view our results in the context of a) the prior extensive research conducted on BVOC fluxes at this very site in previous campaigns, and b) the prior but sparse literature on biogenic light alkene emissions from terrestrial ecosystems. We believe that the important issues raised by the reviewer (such as the relation of alkene emissions to assimilation, respiration rates, etc.) are best addressed in further biogeochemical studies that can better control for these variables, such as leaf or branch chamber type measurements. Doing so here may be too speculative.**

*Page 16, lines 25-27: "We utilized fluxes instead of concentrations to provide a mea-sure of OH reactivity that is independent of elevated concentrations associated with pollution events and more representative of site specific sources." This creates a distorted view of the importance of emissions. The atmospheric OH-reactivity of a VOC at a given concentration ([VOC]) is determined by [VOC] kOH, as mentioned in the previous sentence and the cited literature. In a simplified steady state [VOC] is proportional to the flux and inversely*

*proportional the reactivity (kOH). Consequently,  the overall relative impact of an emission is simply determined by the emission flux.*
*The reactivity determines the temporal and spatial scales at which this happens.  It should also be considered that alkenes also react with ozone, OH reactivity therefore only determines one part of the overall alkene reactivity.*

**After observing the substantial emissions of these compounds, two questions immediately crop up: where are these emissions coming from, and what importance do these emissions have in the atmosphere?  We probed the first question by looking at temporal trends, correlations, understory fluxes and environmental controls.  We probed the second question by comparing the magnitude of light alkene fluxes multiplied by their OH reactivity.  All of the compounds shown in Figure 8 have high OH reactivity but are also long enough lived to measure their fluxes.  Using fluxes instead of concentrations allows us to compare our results with different compounds measured at the site from prior years in other studies.  While intra-canopy OH reactivity may be dominated by some even shorter lived species, this comparison may provide a picture of the OH reactivity in the atmosphere above the canopy.  The point of this exercise is to demonstrate that the light alkene emissions may have a significant impact on the local atmospheric oxidation capacity.  Ozone reactivity could also be illustrated, but we did not feel that it was necessary to demonstrate this point.  Future modeling studies may tackle this question.**

*Page 17, lines 11-12: Based on the uncertainties of averaged fluxes, extrapolation etc. I am not certain that a 20-60% difference is really significant. I also do not understand why the estimate of the seasonal flux average is based on such a simple extrapolation.  The paper presents two more detailed models (gap-filling based on ANN and the T and PAR dependence presented in 5.4) which can be used to calculate averaged fluxes for comparison.*

**In order to compare our results to the one other study of net ecosystem fluxes of light alkenes (Harvard Forest, Goldstein et al., 1996), we needed to extrapolate our July-August fluxes results to cover their integrated period of study from June 1-Oct 31.  Our simple extrapolation essentially assumed that the shoulder months of June and October had emission rates that were half of the mid-season months (i.e., linearly changing from (and to) zero over the course of the shoulder months).  The choice of a simple extrapolation in section 5.3 was because: a) we were not yet convinced of the superiority of any specific model, and b) we did not have a complete season of temperature or PAR measurements from the MEFO tower from which to extrapolate model results (the sonic anemometer was put up early May and removed after the August measurements).  In addition, ANN is a gap-filling model and was not intended to be used as an extrapolation model.**

**In the course of these revisions, we have developed a further understanding and appreciation of the temperature and PAR based model fits.  With the assumption that they can represent shoulder-season fluxes, we now apply the temperature based models to the shoulder season.  To make up for the gap in temperature data at Manitou Forest, we have downloaded weather data for a nearby weather station in Colorado Springs and applied a linear regression with concurrent MEFO measurements to adjust temperatures.  The correction factors were then applied to the temperature data to come up with MEFO**

**temperature for the entire season.  The results demonstrate that a more sophisticated extrapolation of results yields a 40-80% higher emission rate (compared to the 20-60% higher rate from the simple extrapolation).  We now include in section 5.5:  "The choice of which temperature dependent flux response equation to apply varies among different compounds and different studies, as illustrated in Table 3.  In our study, both the light dependent fraction (LDF) and the light-independent fraction (LIDF) equations for temperature response performed better than the PAR response curve.  In addition, the PAR response curve goes to zero as PAR goes to zero, and it appears that emissions of light alkenes occurred at nighttime when PAR equaled zero.  We therefore utilized a combination of the temperature-based equations, scaled by the LDF reported in the MEGAN 2.1 model, to extrapolate flux results to the remainder of the season for which flux measurements were not determined.  Between May 1 and October 31, 2014, the extrapolated seasonal flux yielded an average of 61.5, 51.7, 24.3, and 22.9 $\mu$g m$^{-2}$ h$^{-1}$ for ethene, propene, butene and isoprene, respectively.  For the light alkenes, this represents a 40-80% higher emission rate than that observed over the same season length at Harvard Forest (Goldstein et al., 1996).  This is slightly larger than the simple linear extrapolation described in section 5.3 above and almost identical for isoprene."**

*Combining subchapters 5.4 and 5.5 would allow streamlining the discussion of flux parametrizations and model predictions, for example how well MEGAN 2.1 parame-terizations agree with the measured alkene (and not only isoprene) fluxes.  Based on the current discussion it is not evident that (and why) "Modifying the light and temperature parameterizations for light alkenes in the vegetation emissions model will lead to a corresponding increase in estimated global emissions for these compounds".*

**We chose to keep these sections separate in order to distinguish the modeled curve fits in section 5.4 to their application in the separate MEGAN 2.1 model in section 5.5.  With the addition of the information above in section 5.5, it allows for easier reading.**

*Conclusions:  They are more or less a summary.  I agree with the overall conclusions that the current understanding of alkene emissions from vegetation is insufficient and that alkene emissions from vegetation can be relevant for the chemistry of the atmosphere.  However, based on the many interesting aspects presented and discussed in the paper I am a bit disappointed that the suggestions for tackling open questions and reducing uncertainties is basically a generic "more research needed" approach. Specifically, the very good correlation between ethene and propene fluxes is striking and raises questions about origin and the factors determining the emissions of these two alkenes.*

**We have edited the conclusion paragraph to make it more punchy, with an emphasis on the correlation between ethene and propene and what it means in terms of mutual production mechanisms.  We agree that there were a lot of interesting things to consider in our results, and we appreciate the encouragement to express this more clearly.**

---

## Author Response (AR1)

[revised manuscript text omitted]
 | Measured Flux | ANN Flux[a] | Daytime flux[b] (measured) | Flux under-story | Detection limit |
|---|---|---|---|---|---|---|---|
| | | ppt | µg m$^{-2}$ h$^{-1}$ | µg m$^{-2}$ h$^{-1}$ | µg m$^{-2}$ h$^{-1}$ | µg m$^{-2}$ h$^{-1}$ | µg m$^{-2}$ h$^{-1}$ |
| C$_2$H$_4$ | ethene | 318 [153, 574] | 46.4 [8, 173] | 55.3 [11, 173] | 123 [32, 224] | -33.8 [-63, -1.0] | 4.1 |
| C$_3$H$_6$ | propene | 176 [101, 301] | 35.6 [3, 151] | 43.0 [5, 153] | 94.5 [20, 192] | -40.3 [-62, -5] | 4.7 |
| C$_4$H$_8$ | butene | 52 [29, 103] | 12.0 [0, 59] | 15.6 [1, 61] | 39.1 [15, 80] | -10.4 [-20, -5] | 4.1 |
| C$_5$H$_8$ | isoprene | 115 [31, 297] | 0.6 [-23, 80] | 3.6 [-4, 44] | 17 [-35, 109] | 110 [12, 202] | 3.4 |
| C$_2$H$_2$ | acetylene | 79 [31, 136] | -0.4 [-9, 13] | n/a | -0.2 [-15, 15] | 1.2 [-9, 10] | 13.6 |
| C$_6$H$_6$ | benzene | 43 [25, 68] | -1.6 [-16, 12] | n/a | -3.9 [-28, 16] | -2 [-13, 1] | 5.4 |

[a] Gap filled using artificial neural networks (ANN)
[b] 10-18 MST

**Table 2**. Fitted coefficients for light response flux [with 90% confidence intervals] in Equation 4.

| Response | F(PAR) | | | |
|---|---|---|---|---|
| Compound | F$_{1000}$ [µg m$^{-2}$ hr$^{-1}$] | α (x 10$^{-3}$) | C$_{L1}$ | r$^2$ |
| ethene | 130 | 1.716 [1.097 - 3.379] | 1.1577 [1.0231 - 1.3385] | 0.88 |
| propene | 110.7 | 1.523 [0.5949 - 2.011] | 1.1969 [1.0207 - 1.5171] | 0.83 |
| butene | 37.7 | 1.263 [0.240 - 2.055] | 1.2769 [1.1199 - 2.2605] | 0.86 |
| isoprene | 42.7 | 0.681 [-0.1 - 1.4] | 1.7974 [0.65574 - 3.7002] | 0.8 |
| MBO [a] | | 1.1 | 1.44 | |
| MBO [b] | | 1.1 | 1.37 | |
| MBO [c] | | 1.1 | 1.35 | |

[a] Harley et al., 1998
[b] Schade and Goldstein, 2001
[c] Kaser et al., 2013a

**Table 3.** Fitted coefficients and r$^2$ values of temperature response curves for the light dependent fraction (LDF, Eq. 6) and light independent fraction (LIDF, Eq. 5) for the light alkenes and isoprene. Literature values for the coefficients of other BVOCs are also shown for comparison. For each compound, the LDF currently used in the MEGAN 2.1 model is also indicated. The 90% confidence bounds for the fitted coefficients are in the Supplementary Information.

| | F(T$_{LDF}$) | | | | F(T$_{LIDF}$) | | | LDF[d] |
|---|---|---|---|---|---|---|---|---|
| | Eopt | C$_{T1}$ | C$_{T2}$ | r$^2$ | F$_{ref}$ | β | r$^2$ | |

| Compound | [μg m⁻² hr⁻¹] | | | | [μg m⁻² hr⁻¹] | | | |
|---|---|---|---|---|---|---|---|---|
| Ethene [*] | 228.0 | 165.2 | 168.0 | 0.98 | 316.0 | 0.114 | 0.93 | 0.8 |
| Propene [*] | 410.0 | 116.0 | 148.3 | 0.95 | 326.3 | 0.130 | 0.98 | 0.2 |
| Butene [*] | 231.1 | 139.4 | 146.9 | 0.98 | 115.3 | 0.118 | 0.90 | 0.2 |
| Isoprene [*] | 193.9 | 136.5 | 154.7 | 0.98 | 367.8 | 0.218 | 0.98 | 1.0 |
| MBO [a] | 2200 | 67 | 209 | | | | | 1 |
| MBO [b] | 2000 | 131 | 154 | | | | | 1 |
| MBO [c] | 1800 | 128 | 149 | | | | | 1 |
| methanol [b] | | | | | 7650 | 0.11 | 0.94 | 0.8 |
| methanol [c] | | | | | 940 | 0.13 | 0.81 | 0.8 |
| ethanol [b] | | | | | 1220 | 0.14 | 0.86 | 0.8 |
| ethanol [c] | | | | | 240 | 0.07 | 0.86 | 0.8 |
| acetone [b] | | | | | 590 | 0.11 | 0.98 | 0.2 |
| acetone, propanal [c] | | | | | 630 | 0.15 | 0.92 | 0.2 |
| acetaldehyde [b] | | | | | 360 | 0.13 | 0.92 | 0.8 |
| acetaldehyde [c] | | | | | 330 | 0.12 | 0.85 | 0.8 |
| α-pinene [b] | | | | | 210 | 0.12 | 0.91 | 0.6 |
| monoterpenes [c] | | | | | 500 | 0.12 | 0.85 | 0.4-0.6 |

[*] this study
[a] Harley et al., 1998
[b] Schade and Goldstein 2001
[c] Kaser et al., 2013a.
[d] Guenther et al., 2012

**Figures**

[Figure]

**Figure 1.** The Manitou Experimental Forest Observatory, located in the Front Range of the Rocky Mountains, is shown relative
to the cities of Denver, Boulder, Colorado Springs and Woodland Park in Colorado. Interstate highways 25 and 70 are shown.

[Figure]

**Figure 2.** The Relaxed Eddy Accumulation (REA) system is comprised of: (1) a segregator subsystem and (2) a reservoir subsystem. Sample valves indicated by V, with updraft (up) and downdraft (dn) air sampling valves and bag reservoirs shown.

[Figure]

**Figure 3.** Aerial image of the tower site and the flux footprint (median 90 % recovery) during unstable (blue) and stable (green) atmospheric conditions in this field campaign. Background imagery from Google Earth.

[Figure]

**Figure 4.** Hourly averaged ambient concentrations of alkenes, acetylene and benzene at Manitou Forest. Periods of missing data due to instrumental maintenance or incomplete chromatography.

[Figure]

**Figure 5.** Averaged diurnal patterns of alkene, acetylene and benzene concentrations (red) and their fluxes (blue) with error bars indicating ±1 σ.

[Figure]

**Figure 6.** Net fluxes of (a) ethene, (b) propene, (c) butene and (d) isoprene, based on REA (symbols) and gapfilled with ANN (lines). Measurements of (e) air temperature and cumulative precipitation and (f) PAR and net radiation. Eddy covariance measurements of (g) net $CO_2$ flux (h) water vapor flux and (i) sensible heat flux.

[Figure]

**Figure 7.** Correlation matrix of light alkene, isoprene, acetylene and benzene fluxes (blue), daytime concentrations (red) and night-time concentrations (black). Numbers denote the Pearson correlation coefficient (ρ, top left) and the slope and intercept (bottom right numbers) for the linear fits in plots where ρ > 0.5. Negative fluxes for the light alkenes (1.3% to 2.3% of the light alkene fluxes) are excluded from the plot and the regression statistics; positive fluxes <LDL are not excluded.

[Figure]

**Figure 8.** Daytime averaged molar flux and relative OH reactivity for the major known BVOCs emitted at Manitou Experimental Forest. MBO (not shown) contributes 21 μmol m$^{-2}$ hr$^{-1}$ and 65 % of the OH reactivity.

[Figure]

5 **Figure 9.** Parameterized response curves (solid lines) of alkene fluxes with 10$^{th}$-90$^{th}$ percentile (error bars) for a) the light independent fraction (LIDF) temperature response (Eq 5) bin-averaged into 2 °C classes, and b) the PAR dependent response (Eq. 4) bin-averaged into 200 μmol m$^{-2}$ s$^{-1}$ classes. Response curves are normalized to a flux of 1 at (a) reference temperature of

30 °C and (b) reference PAR of 1000 μmol m$^{-2}$ s$^{-1}$. The light dependent fraction (LDF) temperature response (Eq 6) curve fit is shown in the supplementary text (**Fig. S5**). The response curves in gray and black are for other BVOCs as cited in Tables 2 and 3: [a] Harley et al., 1998; [b] Schade and Goldstein, 2001; [c] Kaser et al., 2013a.

**SUPPLEMENTARY INFORMATION**
**Ethene, propene, butene and isoprene emissions from a ponderosa pine forest measured by Relaxed Eddy Accumulation**

**1. REA measurement quality control**

To test the integrity of air samples collected by REA, several experiments were performed involving variations of sample storage followed by analysis by GC. First, an isoprene standard was measured from the REA bags and from the standard bypassing the REA system; the bag air had a relative error of 2% compared to the standard bypass. Second, carry-over experiments from one bag to the next were performed. Both sets of bags (2 flux periods) were filled with the isoprene standard, followed by GC measurement, sample evacuation and then filling with either (1) zero air (hydrocarbon-free air) or (2) a 50% diluted isoprene standard. For the zero air experiment, a relative isoprene carry-over of 1.4% was detected. For the second experiment, the isoprene measured in the dilution was within 2% of expected.

The transport of air from the sampling inlet to the segregator valves involves a lag time, which needed to be accounted for during conditional sampling. Lag times were experimentally determined in the laboratory using an automated 3-way solenoid pulse valve (MP12-62, Bio-Chem Fluidics Inc., Boonton, NJ, USA) switching between laboratory and $CO_2$-free air and a closed path infrared gas analyzer (Li-6262, LI-COR Biosciences, Lincoln, NE, USA), which was placed downstream of the sampling line. The sampling line lag equals the time between a switch in the valve and an increase/decrease in the $CO_2$ signal. The IRGA response time was measured independently and subtracted from the sampling line lag, to yield a lag of 1.2 seconds at a flow of 315 cc min$^{-1}$ with an inlet line length of 75 cm. The segregator pumping speed (flow rate) was monitored downstream of the neutral line to verify that the flow rate did not change over time; small weekly adjustments of the segregator needle valve were made, as necessary.

For post processing, each hourly REA flux underwent quality control (QC) by applying three tests involving turbulence statistics, REA apparatus performance, and flux footprint analysis. The turbulence test was critical and led to data rejection if the turbulence was poorly developed; it was assessed with tests on the integral turbulence statistics and stationarity (Foken and Wichura, 1996). The second test involved REA apparatus specific checks, with "REA flags" ascribed when (a) more than 5% of the ultrasonic high frequency data were impaired (e.g., due to rain), (b) less than 1.5 L of air was collected in either bag, (c) the Businger-Oncley parameter '$b$' was $\pm 2.5$ standard deviations of the median, (d) there was a small gradient in the proxy scalar ($(\overline{T^+} - \overline{T^-}) < 0.1$ $^o$C), leading to a questionable $b$-value, and/or (e) asynchrony in up- and down-bag sample volume (>15%). None of these flags by themselves were deemed critical failures, but if a majority (3 or more) of these REA flags were present, then the flux measurement failed QC; this situation was rare however. Finally, a non-critical flag was assigned if the flux footprint analysis indicated a possible inhomogeneity (see section 3.7).

In total, 19% of REA data failed QC (were critically flagged), and a further ~12% were marked as "medium quality" based on failing one or two of the REA tests (a-e). Including the footprint test, 47% of REA data were flagged. QC was most sensitive for fluxes close to zero and for apparent uptake (negative fluxes) **(Fig. S1)**. Most of the faulted and flagged fluxes originated from nighttime measurements within a stable boundary layer.

[Figure]

**Fig S1.** Probability density function of ethene fluxes failing QC (dark grey), failing QC or flagged (light gray) and all fluxes (hollow).

[Figure]

5    **Figure S2.** Businger-Oncley parameter (*b*) versus turbulence parameter, calculated for September 1-2, 2014, including the time period of the measured understory fluxes (blue). Nighttime hourly averages (red) mostly fall below the mixing criteria thresholds. 8 of 10 understory flux measurements exceeded the 0.4 threshold which was determined for this site, while the two that did not were early morning fluxes that were near zero.

| | |
|---|---|
| **Deleted:** 6 | |
| **Deleted:** 8 | |

[Figure]

**Figure S3**. Quality-ensured under canopy flux measurements (filled green circles), under canopy fluxes below detection limit (open green circles) and the individual flux detection limits (red lines) overlaid on the hourly averaged fluxes from the above canopy measurements (blue).

[revised manuscript text omitted]
. Seesaw patterns were observed occasionally during the sunrise and sunset transitions, but they were neither systematic (i.e., did not occur regularly) nor consistent (i.e., closer examination shows that fluctuations were not necessarily hourly). In addition, these are periods when ozone concentrations were expected to be low reducing their importance in terms of storage issues. Even under these conditions, negative fluxes were generally not observed.

Another possibility is that the "down" samples descending from the boundary layer could have a slightly higher ozone concentration than the "up" samples rising from the canopy, leading to greater reduction of alkenes in the down bags and hence a small overestimation of calculated emissions. However, the difference in ozone concentrations between up and down bags is likely to be a small percentage of ambient ozone concentrations and hence not likely to influence the overall flux.

During the daytime, ozone photolysis might occur due to sunlight through the portion of Teflon transfer line from the tower that was not covered by foam insulation, which may create OH in the line. However, the relatively high

| Deleted: Th |
| Deleted: e only |
| Deleted: pattern like this occurred |
| Deleted: when |
| Deleted: both fluxes and |
| Deleted: ; e |

concentrations of MBO at the site (ppb level) should act as a built-in scavenger for OH and be its primary loss mechanism (Kim et al., 2010). The product of such a reaction, and for most oxidation reactions of other VOCs, should be oxygenated VOCs rather than ethene, propene or butenes.

**Table S1.** Hydrocarbon compounds measured by GC-FID, concentrations present in the low concentration standard, and instrumental precision determined for the 2 sampling periods.

| Compound | NCAR low concentration standard (ppt) | NOAA standard (CAL018200b) (ppt) | Precision (June 25-29, 2014) | Precision (July 17-August 9, 2014) | Average FID area/ppb response ratio |
|---|---|---|---|---|---|
| Ethane | 549 | 1167 | 8.1% | 10.8% | 18.7 |
| Ethylene | 189 | 1156 | 3.3% | 3.6% | 27.1 |
| Propane | 23 | 1156 | 4.2% | 3.4% | 34.9 |
| Propylene | 59 | 1156 | 6.2% | 7.2% | 43.1 |
| Acetylene | 148 | 1167 | 10.2% | 12.4% | 18.3 |
| i-butane | 76 | 1156 | 8.1% | 8.0% | 42.3 |
| n-butane | 114 | 1156 | 5.5% | 4.3% | 48.6 |
| t-2 butene | -- | 1156 | -- | -- | -- |
| 1-butene | -- | 1112 | -- | -- | -- |
| c-2-butene | -- | 1200 | -- | -- | -- |
| i-pentane | 200 | 1101 | 6.1% | 4.7% | 57.8 |
| n-pentane | 96 | 1134 | 5.9% | 8.8% | 57.7 |
| n-hexane | 47 | 1156 | 10.3% | 6.3% | 55.7 |
| Isoprene | 391 | 1069 | 7.8% | 7.5% | 51.1 |
| Benzene | 86 | 1123 | 6.1% | 6.1% | 69.2 |

**Table S2**. Average concentrations and fluxes of the light alkenes, isoprene, acetylene and benzene at Manitou Forest between June 24-August 9, 2014 and the understory fluxes on September 2, 2014. The statistics include measurements < LDL, and errors reported are 1 sigma. ANN gap-filled fluxes are for the sampling period June 25 to August 9, 2014. The median concentrations with 10th and 90th percentile ranges are reported in Table 1.

| | Compound | Average Concentration ppt | Measured Flux $\mu g\ m^{-2}\ h^{-1}$ | ANN Flux[a] $\mu g\ m^{-2}\ h^{-1}$ | Daytime flux[b] (measured) $\mu g\ m^{-2}\ h^{-1}$ | Flux under-story $\mu g\ m^{-2}\ h^{-1}$ |
|---|---|---|---|---|---|---|
| $C_2H_4$ | ethene | $303.2 \pm 137.7$ | $71.3 \pm 69.5$ | $76.4 \pm 67.1$ | $122.9 \pm 74.2$ | $-31.3 \pm 28.0$ |
| $C_3H_6$ | propene | $181.6 \pm 84.0$ | $59.0 \pm 63.0$ | $63.7 \pm 61.1$ | $103.9 \pm 70.6$ | $-36.6 \pm 26.4$ |
| $C_4H_8$ | butene | $50.6 \pm 29.5$ | $22.8 \pm 29.0$ | $24.8 \pm 26.9$ | $44.4 \pm 32.9$ | $-15.4 \pm 10.1$ |
| $C_5H_8$ | isoprene | $147.5 \pm 97.8$ | $14.3 \pm 49.5$ | $13.2 \pm 30.8$ | $31.9 \pm 66.4$ | $108 \pm 101.7$ |
| $C_2H_2$ | acetylene | $85.7 \pm 44.4$ | $0.9 \pm 13.5$ | n/a | $1.5 \pm 18.4$ | $1.0 \pm 9.7$ |

| $C_6H_6$ | benzene | 43.8 ± 16.7 | -2.6 ± 17.1 | n/a | -5.9 ± 21.4 | -5.1 ± 6.6 |

[a] Gap filled using artificial neural networks (ANN)

[b] 10-18 MST

**Table S3.** Supplement to Table 2 in the manuscript. Fitted coefficients with their 10[th] and 90[th] percentiles of temperature response curves for the light dependent fraction (LDF, Eq. 6) and light independent fraction (LIDF, Eq. 5) for the light alkenes and isoprene.

| | $F(T_{LDF})$ Coefficients (Eq. 6) | | | $F(T_{LIDF})$ Coefficients (Eq. 5) | |
|---|---|---|---|---|---|
| | $E_{opt}$ | $C_{T1}$ | $C_{T2}$ | $F_{ref}$ | $\beta$ |
| | [$\mu g\ m^{-2}\ hr^{-1}$] | | | [$\mu g\ m^{-2}\ hr^{-1}$] | |
| ethene | 228 (186.8 - 269.2) | 165.2 (124.6 - 205.8) | 168.0 (131.0 - 205.0) | 316.0 (242.3 - 389.8) | 0.114 (0.089 - 0.140) |
| propene | 410 (168.9 - 651.1) | 116.0 (83.2 - 148.7) | 148.3 (119.5 - 177.0) | 326.3 (286.0 - 366.5) | 0.130 (0.116 - 0.144) |
| butene | 231.1 (173.9 - 288.3) | 139.4 (103.4 - 175.3) | 146.9 (120.2 - 173.6) | 115.3 (59.3 171.4) | 0.118 (0.063 - 0.173) |
| isoprene | 193.9 (17.9 - 369.9) | 136.5 (65.0 - 208.0) | 154.7 (132.9 - 176.4) | 367.8 (287.2 - 454.4) | 0.218 (0.183 - 0.252) |

[Figure]

**Figure S5.** Parameterized response curves (solid lines) of alkene fluxes bin-averaged in 2K classes (circles) with 10[th]-90[th] percentile (error bars) for the light dependent fraction (LDF) temperature response (Eq 6). Response curves have been normalized to a flux of 1 at a reference temperature of 312K (39 °C). The response curves in gray, green and black are for MBO as cited in Table 3: [a] Harley et al., 1998; [b] Schade and Goldstein, 2001; [c] Kaser et al., 2013a.